# Provably Robust Deep Learning via Adversarially Trained Smoothed Classifiers

**Hadi Salman**[†], **Greg Yang**[§], **Jerry Li**,
**Pengchuan Zhang**[∗], **Huan Zhang**[∗], **Ilya Razenshteyn**[∗], **Sébastien Bubeck**[∗]
Microsoft Research AI
{hadi.salman, gregyang, jerrl,
penzhan, t-huzhan, ilyaraz, sebubeck }@microsoft.com

## Abstract

Recent works have shown the effectiveness of *randomized smoothing* as a scalable technique for building neural network-based classifiers that are provably robust to $\ell_2$-norm adversarial perturbations. In this paper, we employ adversarial training to improve the performance of randomized smoothing. We design an adapted attack for smoothed classifiers, and we show how this attack can be used in an adversarial training setting to boost the *provable* robustness of smoothed classifiers. We demonstrate through extensive experimentation that our method consistently outperforms all existing provably $\ell_2$-robust classifiers by a significant margin on ImageNet and CIFAR-10, establishing the state-of-the-art for provable $\ell_2$-defenses. Moreover, we find that pre-training and semi-supervised learning boost adversarially trained smoothed classifiers even further. Our code and trained models are available at http://github.com/Hadisalman/smoothing-adversarial[2].

## 1   Introduction

Neural networks have been very successful in tasks such as image classification and speech recognition, but have been shown to be extremely brittle to small, adversarially-chosen perturbations of their inputs [33, 14]. A classifier (e.g., a neural network), which correctly classifies an image $x$, can be fooled by an adversary to misclassify $x + \delta$ where $\delta$ is an adversarial perturbation so small that $x$ and $x + \delta$ are indistinguishable for the human eye. Recently, many works have proposed heuristic defenses intended to train models robust to such adversarial perturbations. However, most of these defenses were broken using more powerful adversaries [4, 2, 35]. This encouraged researchers to develop defenses that lead to *certifiably robust* classifiers, i.e., whose predictions for most of the test examples $x$ can be verified to be constant within a neighborhood of $x$ [39, 27]. Unfortunately, these techniques do not immediately scale to large neural networks that are used in practice.

To mitigate this limitation of prior certifiable defenses, a number of papers [21, 22, 6] consider the *randomized smoothing* approach, which transforms any classifier $f$ (e.g., a neural network) into a new *smoothed* classifier $g$ that has certifiable $\ell_2$-norm robustness guarantees. This transformation works as follows.

Let $f$ be an arbitrary base classifier which maps inputs in $\mathbb{R}^d$ to classes in $\mathcal{Y}$. Given an input $x$, the smoothed classifier $g(x)$ labels $x$ as having class $c$ which is the most likely to be returned by the base classifier $f$ when fed a noisy corruption $x + \delta$, where $\delta \sim \mathcal{N}(x, \sigma^2 I)$ is a vector sampled according to an isotropic Gaussian distribution.

As shown in [6], one can derive certifiable robustness for such smoothed classifiers via the Neyman-Pearson lemma. They demonstrate that for $\ell_2$ perturbations, randomized smoothing outperforms

---

[∗]Reverse alphabetical order. [†]Work done as part of the Microsoft AI Residency Program. [§]Primary mentor.
[2]Please see http://arxiv.org/abs/1906.04584 for the full and most recent version of this paper.

Table 1: Certified top-1 accuracy of our best ImageNet classifiers at various $\ell_2$ radii.

| $\ell_2$ RADIUS (IMAGENET) | 0.5 | 1.0 | 1.5 | 2.0 | 2.5 | 3.0 | 3.5 |
|---|---|---|---|---|---|---|---|
| COHEN ET AL. [6] (%) | 49 | 37 | 29 | 19 | 15 | 12 | 9 |
| OURS (%) | **56** | **45** | **38** | **28** | **26** | **20** | **17** |

Table 2: Certified top-1 accuracy of our best CIFAR-10 classifiers at various $\ell_2$ radii.

| $\ell_2$ RADIUS (CIFAR-10) | 0.25 | 0.5 | 0.75 | 1.0 | 1.25 | 1.5 | 1.75 | 2.0 | 2.25 |
|---|---|---|---|---|---|---|---|---|---|
| COHEN ET AL. [6] (%) | 61 | 43 | 32 | 22 | 17 | 13 | 10 | 7 | 4 |
| OURS (%) | 73 | 58 | 48 | 38 | 33 | 29 | 24 | 18 | 16 |
| + PRE-TRAINING (%) | 80 | 62 | **52** | 38 | **34** | **30** | 25 | **19** | 16 |
| + SEMI-SUPERVISION (%) | 80 | **63** | **52** | **40** | **34** | 29 | 25 | **19** | **17** |
| + BOTH(%) | **81** | **63** | **52** | 37 | 33 | 29 | **25** | 18 | 16 |

other certifiably robust classifiers that have been previously proposed. It is scalable to networks with any architecture and size, which makes it suitable for building robust real-world neural networks.

**Our contributions**   In this paper, we employ adversarial training to substantially improve on the previous certified robustness results[3] of randomized smoothing [21, 22, 6]. We present, for the first time, a direct attack for smoothed classifiers. We then demonstrate how to use this attack to adversarially train smoothed models with not only boosted empirical robustness but also **substantially improved certifiable robustness** using the certification method of Cohen et al. [6].

We demonstrate that our method outperforms *all* existing provably $\ell_2$-robust classifiers by a significant margin on ImageNet and CIFAR-10, establishing the state-of-the-art for provable $\ell_2$-defenses. For instance, our Resnet-50 ImageNet classifier achieves 56% provable top-1 accuracy (compared to the best previous provable accuracy of 49%) under adversarial perturbations with $\ell_2$ norm less than 127/255. Similarly, our Resnet-110 CIFAR-10 smoothed classifier achieves up to 16% improvement over previous state-of-the-art, and by combining our technique with pre-training [17] and semi-supervised learning [5], we boost our results to up to 22% improvement over previous state-of-the-art. Our main results are reported in Tables 1 and 2 for ImageNet and CIFAR-10. See Tables 16 and 17 in Appendix G for the standard accuracies corresponding to these results.

Finally, we provide an alternative, but more concise, proof of the tight robustness guarantee of Cohen et al. [6] by casting this as a *nonlinear* Lipschitz property of the smoothed classifier. See appendix A for the complete proof.

## 2   Our techniques

Here we describe our techniques for adversarial attacks and training on smoothed classifiers. We first require some background on randomized smoothing classifiers. For a more detailed description of randomized smoothing, see Cohen et al. [6].

### 2.1   Background on randomized smoothing

Consider a classifier $f$ from $\mathbb{R}^d$ to classes $\mathcal{Y}$. Randomized smoothing is a method that constructs a new, *smoothed* classifier $g$ from the *base* classifier $f$. The smoothed classifier $g$ assigns to a query point $x$ the class which is most likely to be returned by the base classifier $f$ under isotropic Gaussian noise perturbation of $x$, i.e.,

$$g(x) = \arg\max_{c \in \mathcal{Y}} \mathbb{P}(f(x + \delta) = c) \quad \text{where } \delta \sim \mathcal{N}(0, \sigma^2 I) . \tag{1}$$

The noise level $\sigma^2$ is a hyperparameter of the smoothed classifier $g$ which controls a robustness/accuracy tradeoff. Equivalently, this means that $g(x)$ returns the class $c$ whose decision region $\{x' \in \mathbb{R}^d : f(x') = c\}$ has the largest measure under the distribution $\mathcal{N}(x, \sigma^2 I)$. Cohen et al. [6]

recently presented a tight robustness guarantee for the smoothed classifier $g$ and gave Monte Carlo algorithms for certifying the robustness of $g$ around $x$ or predicting the class of $x$ using $g$, that succeed with high probability.

**Robustness guarantee for smoothed classifiers**    The robustness guarantee presented by [6] uses the Neyman-Pearson lemma, and is as follows: suppose that when the base classifier $f$ classifies $\mathcal{N}(x, \sigma^2 I)$, the class $c_A$ is returned with probability $p_A = \mathbb{P}(f(x + \delta) = c_A)$, and the "runner-up" class $c_B$ is returned with probability $p_B = \max_{c \neq c_A} \mathbb{P}(f(x + \delta) = c)$. The smoothed classifier $g$ is robust around $x$ within the radius

$$R = \frac{\sigma}{2} \left( \Phi^{-1}(p_A) - \Phi^{-1}(p_B) \right), \tag{2}$$

where $\Phi^{-1}$ is the inverse of the standard Gaussian CDF. It is not clear how to compute $p_A$ and $p_B$ exactly (if $f$ is given by a deep neural network for example). Monte Carlo sampling is used to estimate some $\underline{p_A}$ and $\overline{p_B}$ for which $\underline{p_A} \leq p_A$ and $\overline{p_B} \geq p_B$ with arbitrarily high probability over the samples. The result of (2) still holds if we replace $p_A$ with $\underline{p_A}$ and $p_B$ with $\overline{p_B}$.

This guarantee can in fact be obtained alternatively by explicitly computing the Lipschitz constant of the smoothed classifier, as we do in Appendix A.

## 2.2  SMOOTHADV: Attacking smoothed classifiers

We now describe our attack against smoothed classifiers. To do so, it will first be useful to describe smoothed classifiers in a more general setting. Specifically, we consider a generalization of (1) to *soft* classifiers, namely, functions $F : \mathbb{R}^d \to P(\mathcal{Y})$, where $P(\mathcal{Y})$ is the set of probability distributions over $\mathcal{Y}$. Neural networks typically learn such soft classifiers, then use the argmax of the soft classifier as the final hard classifier. Given a soft classifier $F$, its associated *smoothed* soft classifier $G : \mathbb{R}^n \to P(\mathcal{Y})$ is defined as

$$G(x) = \left( F * \mathcal{N}(0, \sigma^2 I) \right)(x) = \mathop{\mathbb{E}}_{\delta \sim \mathcal{N}(0, \sigma^2 I)} [F(x + \delta)]. \tag{3}$$

Let $f(x)$ and $F(x)$ denote the hard and soft classifiers learned by the neural network, respectively, and let $g$ and $G$ denote the associated smoothed hard and smoothed soft classifiers. Directly finding adversarial examples for the smoothed *hard* classifier $g$ is a somewhat ill-behaved problem because of the argmax, so we instead propose to *find adversarial examples for the smoothed soft classifier $G$*. Empirically we found that doing so will also find good adversarial examples for the smoothed hard classifier. More concretely, given a labeled data point $(x, y)$, we wish to find a point $\hat{x}$ which maximizes the loss of $G$ in an $\ell_2$ ball around $x$ for some choice of loss function. As is canonical in the literature, we focus on the cross entropy loss $\ell_{\mathrm{CE}}$. Thus, given a labeled data point $(x, y)$ our (ideal) adversarial perturbation is given by the formula:

$$\hat{x} = \mathop{\arg\max}_{\|x' - x\|_2 \leq \epsilon} \ell_{\mathrm{CE}}(G(x'), y)$$

$$= \mathop{\arg\max}_{\|x' - x\|_2 \leq \epsilon} \left( -\log \mathop{\mathbb{E}}_{\delta \sim \mathcal{N}(0, \sigma^2 I)} \left[ (F(x' + \delta))_y \right] \right). \tag{$\mathcal{S}$}$$

We will refer to ($\mathcal{S}$) as the SMOOTHADV objective. The SMOOTHADV objective is highly non-convex, so as is common in the literature, we will optimize it via projected gradient descent (PGD), and variants thereof. It is hard to find exact gradients for ($\mathcal{S}$), so in practice we must use some estimator based on random Gaussian samples. There are a number of different natural estimators for the derivative of the objective function in ($\mathcal{S}$), and the choice of estimator can dramatically change the performance of the attack. For more details, see Section 3.

We note that ($\mathcal{S}$) should not be confused with the similar-looking objective

$$\hat{x}_{\mathrm{wrong}} = \mathop{\arg\max}_{\|x' - x\|_2 \leq \epsilon} \left( \mathop{\mathbb{E}}_{\delta \sim \mathcal{N}(0, \sigma^2 I)} \left[ \ell_{\mathrm{CE}}(F(x' + \delta), y) \right] \right)$$

$$= \mathop{\arg\max}_{\|x' - x\|_2 \leq \epsilon} \left( \mathop{\mathbb{E}}_{\delta \sim \mathcal{N}(0, \sigma^2 I)} \left[ -\log \left( F(x' + \delta) \right)_y \right] \right), \tag{4}$$

as suggested in section G.3 of [6]. There is a subtle, but very important, distinction between ($\mathcal{S}$) and (4). Conceptually, solving (4) corresponds to finding an adversarial example of $F$ that is robust to

Gaussian noise. In contrast, ($\mathcal{S}$) is directly attacking the smoothed model i.e. trying to find adversarial examples that decrease the probability of correct classification of the smoothed soft classifier $G$. From this point of view, ($\mathcal{S}$) is the right optimization problem that should be used to find adversarial examples of $G$. This distinction turns out to be crucial in practice: empirically, Cohen et al. [6] found attacks based on (4) not to be effective.

Interestingly, for a large class of classifiers, including neural networks, one can alternatively derive the objective ($\mathcal{S}$) from an optimization perspective, by attempting to directly find adversarial examples to the smoothed hard classifier that the neural network provides. While they ultimately yield the same objective, this perspective may also be enlightening, and so we include it in Appendix B.

### 2.3 Adversarial training using SMOOTHADV

We now wish to use our new attack to boost the adversarial robustness of smoothed classifiers. We do so using the well-studied *adversarial training* framework [20, 25]. In adversarial training, given a current set of model weights $w_t$ and a labeled data point $(x_t, y_t)$, one finds an adversarial perturbation $\hat{x}_t$ of $x_t$ for the current model $w_t$, and then takes a gradient step for the model parameters, evaluated at the point $(\hat{x}_t, y_t)$. Intuitively, this encourages the network to learn to minimize the worst-case loss over a neighborhood around the input.

At a high level, we propose to instead do adversarial training using an adversarial example *for the smoothed classifier*. We combine this with the approach suggested in Cohen et al. [6], and train at Gaussian perturbations of this adversarial example. That is, given current set of weights $w_t$ and a labeled data point $(x_t, y_t)$, we find $\hat{x}_t$ as a solution to ($\mathcal{S}$), and then take a gradient step for $w_t$ based at gaussian perturbations of $\hat{x}_t$. In contrast to standard adversarial training, we are training the base classifier so that its associated smoothed classifier minimizes worst-case loss in a neighborhood around the current point. For more details of our implementation, see Section 3.2. *We emphasize that although we are training using adversarial examples for the smoothed soft classifier, in the end we certify the robustness of the smoothed hard classifier we obtain after training.*

We make two important observations about our method. First, adversarial training is an empirical defense, and typically offers no provable guarantees. However, we demonstrate that by combining our formulation of adversarial training with randomized smoothing, we are able to substantially boost the certifiable robust accuracy of our smoothed classifiers. Thus, while adversarial training using SMOOTHADV is still ultimately a heuristic, and offers no provable robustness by itself, the smoothed classifier that we obtain using this heuristic has strong certifiable guarantees.

Second, we found empirically that to obtain strong certifiable numbers using randomized smoothing, it is *insufficient* to use standard adversarial training on the *base* classifier. While such adversarial training does indeed offer good empirical robust accuracy, the resulting classifier is not optimized for randomized smoothing. In contrast, our method specifically finds base classifiers whose smoothed counterparts are robust. As a result, the certifiable numbers for standard adversarial training are noticeably worse than those obtained using our method. See Appendix C.1 for an in-depth comparison.

## 3 Implementing SMOOTHADV via first order methods

As mentioned above, it is difficult to optimize the SMOOTHADV objective, so we will approximate it via first order methods. We focus on two such methods: the well-studied *projected gradient descent (PGD)* method [20, 25], and the recently proposed *decoupled direction and norm (DDN)* method [29] which achieves $\ell_2$ robust accuracy competitive with PGD on CIFAR-10.

The main task when implementing these methods is to, given a data point $(x, y)$, compute the gradient of the objective function in ($\mathcal{S}$) with respect to $x'$. If we let $J(x') = \ell_{CE}(G(x'), y)$ denote the objective function in ($\mathcal{S}$), we have

$$\nabla_{x'} J(x') = \nabla_{x'} \left( -\log \mathbb{E}_{\delta \sim \mathcal{N}(0, \sigma^2 I)} [F(x' + \delta)_y] \right) . \tag{5}$$

However, it is not clear how to evaluate (5) exactly, as it takes the form of a complicated high dimensional integral. Therefore, we will use Monte Carlo approximations. We sample i.i.d. Gaussians $\delta_1, \ldots, \delta_m \sim \mathcal{N}(0, \sigma^2 I)$, and use the plug-in estimator for the expectation:

$$\nabla_{x'} J(x') \approx \nabla_{x'} \left( -\log \left( \frac{1}{m} \sum_{i=1}^{m} F(x' + \delta_i)_y \right) \right) . \tag{6}$$

**Pseudocode 1:** SMOOTHADV-ersarial Training

---

**function** TRAINMINIBATCH($(x^{(1)}, y^{(1)}), (x^{(2)}, y^{(2)}), \ldots, (x^{(B)}, y^{(B)})$)
    ATTACKER $\leftarrow$ (SMOOTHADV$_{\text{PGD}}$ or SMOOTHADV$_{\text{DDN}}$)
    Generate noise samples $\delta_i^{(j)} \sim \mathcal{N}(0, \sigma^2 I)$ for $1 \leq i \leq m, 1 \leq j \leq B$
    $L \leftarrow []$    *# List of adversarial examples for training*
    **for** $1 \leq j \leq B$ **do**
        $\hat{x}^{(j)} \leftarrow x^{(j)}$    *# Adversarial example*
        **for** $1 \leq k \leq T$ **do**
            Update $\hat{x}^{(j)}$ according to the $k$-th step of ATTACKER, where we use
            the noise samples $\delta_1^{(j)}, \delta_2^{(j)}, \ldots, \delta_m^{(j)}$ to estimate a gradient of the loss of the smoothed
            model according to (6)
            *# We are reusing the same noise samples between different steps of the attack*
        **end**
        Append $((\hat{x}^{(j)} + \delta_1^{(j)}, y^{(j)}), (\hat{x}^{(j)} + \delta_2^{(j)}, y^{(j)}), \ldots, (\hat{x}^{(j)} + \delta_m^{(j)}, y^{(j)}))$ to $L$
        *# Again, we are reusing the same noise samples for the augmentation*
    **end**
    Run backpropagation on $L$ with an appropriate learning rate

---

It is not hard to see that if $F$ is smooth, this estimator will converge to (5) as we take more samples. In practice, if we take $m$ samples, then to evaluate (6) on all $m$ samples requires evaluating the network $m$ times. This becomes expensive for large $m$, especially if we want to plug this into the adversarial training framework, which is already slow. Thus, when we use this for adversarial training, we use $m_{\text{train}} \in \{1, 2, 4, 8\}$. When we run this attack to evaluate the *empirical* adversarial accuracy of our models, we use substantially larger choices of $m$, specifically, $m_{\text{test}} \in \{1, 4, 8, 16, 64, 128\}$. Empirically we found that increasing $m$ beyond 128 did not substantially improve performance.

While this estimator does converge to the true gradient given enough samples, note that it is not an unbiased estimator for the gradient. Despite this, we found that using (6) performs very well in practice. Indeed, using (6) yields our strongest empirical attacks, as well as our strongest certifiable defenses when we use this attack in adversarial training. In the remainder of the paper, we let SMOOTHADV$_{\text{PGD}}$ denote the PGD attack with gradient steps given by (6), and similarly we let SMOOTHADV$_{\text{DDN}}$ denote the DDN attack with gradient steps given by (6).

### 3.1 An unbiased, gradient free method

We note that there is an alternative way to optimize ($\mathcal{S}$) using first order methods. Notice that the logarithm in ($\mathcal{S}$) does not change the argmax, and so it suffices to find a minimizer of $G(x')_y$ subject to the $\ell_2$ constraint. We then observe that

$$\nabla_{x'}(G(x')_y) = \mathbb{E}_{\delta \sim \mathcal{N}(0, \sigma^2 I)} \left[ \nabla_{x'} F(x' + \delta)_y \right] \overset{(a)}{=} \mathbb{E}_{\delta \sim \mathcal{N}(0, \sigma^2 I)} \left[ \frac{\delta}{\sigma^2} \cdot F(x' + \delta)_y \right] . \tag{7}$$

The equality (a) is known as *Stein's lemma* [32], although we note that something similar can be derived for more general distributions. There is a natural unbiased estimator for (7): sample i.i.d. gaussians $\delta_1, \ldots, \delta_m \sim \mathcal{N}(0, \sigma^2 I)$, and form the estimator $\nabla_{x'}(G(x')_y) \approx \frac{1}{m} \sum_{i=1}^{m} \frac{\delta_i}{\sigma^2} \cdot F(x' + \delta_i)_y$ . This estimator has a number of nice properties. As mentioned previously, it is an unbiased estimator for (7), in contrast to (6). It also requires no computations of the gradient of $F$; if $F$ is a neural network, this saves both time and memory by not storing preactivations during the forward pass. Finally, it is very general: the derivation of (7) actually holds even if $F$ is a hard classifier (or more precisely, the one-hot embedding of a hard classifier). In particular, this implies that this technique can even be used to directly find adversarial examples of the smoothed hard classifier.

Despite these appealing features, in practice we find that this attack is quite weak. We speculate that this is because the variance of the gradient estimator is too high. For this reason, in the empirical evaluation we focus on attacks using (6), but we believe that investigating this attack in practice is an interesting direction for future work. See Appendix C.6 for more details.

### 3.2 Implementing adversarial training for smoothed classifiers

We incorporate adversarial training into the approach of Cohen et al. [6] changing as few moving parts as possible in order to enable a direct comparison. In particular, we use the same network architectures, batch size, and learning rate schedule. For CIFAR-10, we change the number of epochs,

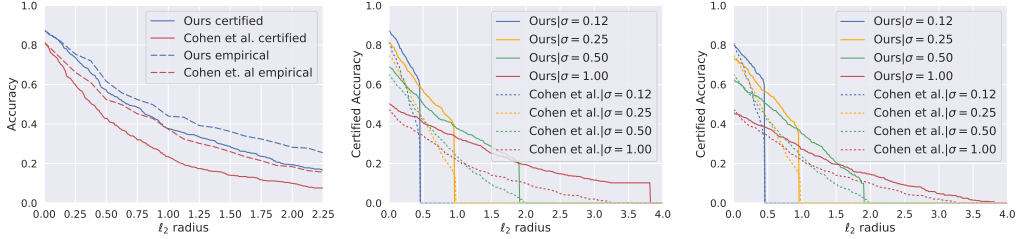

Figure 1: Comparing our SMOOTHADV-ersarially trained CIFAR-10 classifiers vs Cohen et al. [6]. **(Left)** Upper envelopes of certified accuracies over all experiments. **(Middle)** Upper envelopes of certified accuracies per $\sigma$. **(Right)** Certified accuracies of one representative model per $\sigma$. Details of each model used to generate these plots and their certified accuracies are in Tables 7-15 in Appendix G.

but for ImageNet, we leave it the same. We discuss more of these specifics in Appendix D, and here we describe how to perform adversarial training on a single mini-batch. The algorithm is shown in Pseudocode 1, with the following parameters: $B$ is the mini-batch size, $m$ is the number of noise samples used for gradient estimation in (6) as well as for Gaussian noise data augmentation, and $T$ is the number of steps of an attack[4].

# 4 Experiments

We primarily compare with Cohen et al. [6] as it was shown to outperform all other scalable provable $\ell_2$ defenses by a wide margin. As our experiments will demonstrate, our method consistently and significantly outperforms Cohen et al. [6] even further, establishing the state-of-the-art for provable $\ell_2$-defenses. We run experiments on ImageNet [8] and CIFAR-10 [19]. We use the same base classifiers $f$ as Cohen et al. [6]: a ResNet-50 [16] on ImageNet, and ResNet-110 on CIFAR-10. Other than the choice of attack (SMOOTHADV$_{\text{PGD}}$ or SMOOTHADV$_{\text{DDN}}$) for adversarial training, our experiments are distinguished based on five main hyperparameters:

$$
\begin{aligned}
\epsilon &= \text{maximum allowed } \ell_2 \text{ perturbation of the input} \\
T &= \text{number of steps of the attack} \\
\sigma &= \text{std. of Gaussian noise data augmentation during training and certification} \\
m_{\text{train}} &= \text{number of noise samples used to estimate (6) during training} \\
m_{\text{test}} &= \text{number of noise samples used to estimate (6) during evaluation}
\end{aligned}
\qquad (\diamondsuit)
$$

Given a smoothed classifier $g$, we use the same prediction and certification algorithms, PREDICT and CERTIFY, as [6]. Both algorithms sample base classifier predictions under Gaussian noise. PREDICT outputs the majority vote if the vote count passes a binomial hypothesis test, and abstains otherwise. CERTIFY certifies the majority vote is robust if the fraction of such votes is higher by a calculated margin than the fraction of the next most popular votes, and abstains otherwise. For details of these algorithms, we refer the reader to [6].

The **certified accuracy** at radius $r$ is defined as the fraction of the test set which $g$ classifies correctly (without abstaining) and certifies robust at an $\ell_2$ radius $r$. Unless otherwise specified, we use the same $\sigma$ for certification as the one used for training the base classifier $f$. Note that $g$ is a randomized smoothing classifier, so this reported accuracy is *approximate*, but can get arbitrarily close to the *true* certified accuracy as the number of samples of $g$ increases (see [6] for more details). Similarly, the **empirical accuracy** is defined as the fraction of the $\ell_2$ SMOOTHADV-ersarially attacked test set which $g$ classifies correctly (without abstaining). Both PREDICT and CERTIFY have a parameter $\alpha$ defining the failure rate of these algorithms. Throughout the paper, we set $\alpha = 0.001$ (similar to [6]), which means there is at most a 0.1% chance that PREDICT *does not* return the most probable class under the smoothed classifier $g$, or that CERTIFY falsely certifies a non-robust input.

## 4.1 SMOOTHADV-ersarial training

To assess the effectiveness of our method, we learn a smoothed classifier $g$ that is adversarial trained using ($\mathcal{S}$). Then we compute the *certified accuracies*[5] over a range of $\ell_2$ radii $r$. Tables 1 and 2

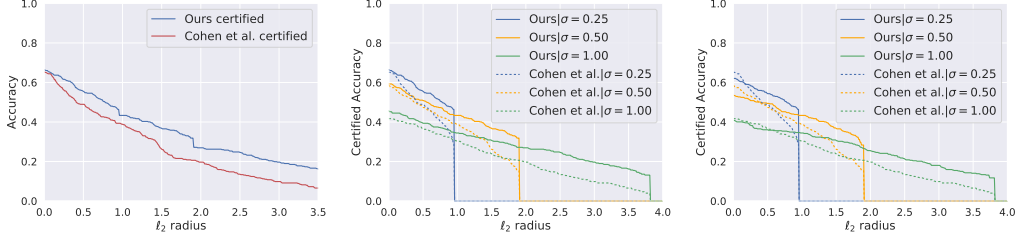

Figure 2: Comparing our SMOOTHADV-ersarially trained ImageNet classifiers vs Cohen et al. [6]. Subfigure captions are same as Fig. 1. Details of each model used to generate these plots and their certified accuracies are in Table 6 in Appendix G.

Table 3: Certified $\ell_\infty$ robustness at a radius of $\frac{2}{255}$ on CIFAR-10. Note that our models and Carmon et al. [5]'s give accuracies with high probability (W.H.P).

| MODEL | $\ell_\infty$ ACC. AT 2/255 | STANDARD ACC. |
|---|---|---|
| OURS (%) | **68.2** (W.H.P) | **86.2** (W.H.P) |
| CARMON ET AL. [5] (%) | $63.8 \pm 0.5$ (W.H.P) | $80.7 \pm 0.3$ (W.H.P) |
| WONG AND KOLTER [39] (SINGLE) (%) | 53.9 | 68.3 |
| WONG AND KOLTER [39] (ENSEMBLE) (%) | 63.6 | 64.1 |
| IBP [15] (%) | 50.0 | 70.2 |

report the certified accuracies using our method compared to [6]. For all radii, we outperform the certified accuracies of [6] by a significant margin on both ImageNet and CIFAR-10. These results are elaborated below.

**For CIFAR-10** Fig. 1(left) plots the upper envelope of the certified accuracies that we get by choosing the best model for each radius over a grid of hyperparameters. This grid consists of $m_{train} \in \{1, 2, 4, 8\}$, $\sigma \in \{0.12, 0.25, 0.5, 1.0\}$, $\epsilon \in \{0.25, 0.5, 1.0, 2.0\}$ (see $\diamond$ for explanation), and one of the following attacks {SMOOTHADV$_{PGD}$, SMOOTHADV$_{DDN}$} with $T \in \{2, 4, 6, 8, 10\}$ steps. The certified accuracies of each model can be found in Tables 7-15 in Appendix G. These results are compared to those of Cohen et al. [6] by plotting their reported certified accuracies. Fig. 1(left) also plots the corresponding empirical accuracies using SMOOTHADV$_{PGD}$ with $m_{test} = 128$. Note that *our **certified** accuracies are higher than the **empirical** accuracies of Cohen et al. [6].*

Fig. 1(middle) plots our vs [6]'s best models for varying noise level $\sigma$. Fig. 1(right) plots a representative model for each $\sigma$ from our adversarially trained models. Observe that we outperform [6] in all three plots.

**For ImageNet** The results are summarized in Fig. 2, which is similar to Fig. 1 for CIFAR-10, with the difference being the set of smoothed models we certify. This set includes smoothed models trained using $m_{\text{train}} = 1$, $\sigma \in \{0.25, 0.5, 1.0\}$, $\epsilon \in \{0.5, 1.0, 2.0, 4.0\}$, and one of the following attacks {1-step SMOOTHADV$_{PGD}$, 2-step SMOOTHADV$_{DDN}$}. Again, our models outperform those of Cohen et al. [6] overall and per $\sigma$ as well. The certified accuracies of each model can be found in Table 6 in Appendix G.

We point out, as mentioned by Cohen et al. [6], that $\sigma$ controls a robustness/accuracy trade-off. When $\sigma$ is low, small radii can be certified with high accuracy, but large radii cannot be certified at all. When $\sigma$ is high, larger radii can be certified, but smaller radii are certified at a lower accuracy. This can be observed in the middle and the right plots of Fig. 1 and 2.

**Effect on clean accuracy** Training smoothed classifers using SMOOTHADV as shown improves upon the certified accuracy of Cohen et al. [6] for each $\sigma$, although this comes with the well-known effect of adversarial training in decreasing the standard accuracy, so we sometimes see small drops in the accuracy at $r = 0$, as observed in Fig. 1(right) and 2(right).

$\ell_2$ **to** $\ell_\infty$ **certified defense** Since the $\ell_2$ ball of radius $\sqrt{d}$ contains the $\ell_\infty$ unit ball in $\mathbb{R}^d$, a model robust against $\ell_2$ perturbation of radius $r$ is also robust against $\ell_\infty$ perturbation of norm $r/\sqrt{d}$. Via this naive conversion, we find our $\ell_2$-robust models enjoy non-trivial $\ell_\infty$ certified robustness.

In Table 3, we report the best[6] $\ell_\infty$ certified accuracy that we get on CIFAR-10 at a radius of 2/255 (implied by the $\ell_2$ certified accuracy at a radius of $0.435 \approx 2\sqrt{3 \times 32^2}/255$). We exceed previous state-of-the-art in certified $\ell_\infty$ defenses by at least 3.9%. We obtain similar results for ImageNet certified $\ell_\infty$ defenses at a radius of 1/255 where we exceed the previous state-of-the-art by 8.2%; details are in appendix F.

**Additional experiments and observations**  We compare the effectiveness of smoothed classifiers when they are trained SMOOTHADV-versarially vs. when their *base* classifier is trained via standard adversarial training (we will refer to the latter as *vanilla adversarial training*). As expected, because the training objective of SMOOTHADV-models aligns with the actual certification objective, those models achieve noticeably more certified robustness over all radii compared to smoothed classifiers resulting from vanilla adversarial training. We defer the results and details to Appendix C.1.

Furthermore, SMOOTHADV requires the evaluation of (6) as discussed in Section 3. We analyze in Appendix C.2 how the number of Gaussian noise samples $m_{\text{train}}$, used in (6) to find adversarial examples, affects the robustness of the resulting smoothed models. As expected, we observe that models trained with higher $m_{\text{train}}$ tend to have higher certified accuracies.

Finally, we analyze the effect of the maximum allowed $\ell_2$ perturbation $\epsilon$ used in SMOOTHADV on the robustness of smoothed models in Appendix C.3. We observe that as $\epsilon$ increases, the certified accuracies for small $\ell_2$ radii decrease, but those for large $\ell_2$ radii increase, which is expected.

## 4.2 More Data for Better Provable Robustness

We explore using more data to improve the robustness of smoothed classifiers. Specifically, we pursue two ideas: 1) *pre-training* similar to [17], and 2) *semi-supervised learning* as in [5].

**Pre-training**  Hendrycks et al. [17] recently showed that using pre-training can improve the adversarial robustness of classifiers, and achieved state-of-the-art results for *empirical* $l_\infty$ defenses on CIFAR-10 and CIFAR-100. We employ this within our framework; we pretrain smoothed classifiers on ImageNet, then fine-tune them on CIFAR-10. Details can be found in Appendix E.1.

**Semi-supervised learning**  Carmon et al. [5] recently showed that using unlabelled data can improve the adversarial robustness as well. They employ a simple, yet effective, semi-supervised learning technique called *self-training* to improve the robustness of CIFAR-10 classifiers. We employ this idea in our framework and we train our CIFAR-10 smoothed classifiers via self-training using the unlabelled dataset used in Carmon et al. [5]. Details can be found in Appendix E.2.

We further experiment with combining semi-supervised learning and pre-training, and the details are in Appendix E.3. We observe consistent improvement in the certified robustness of our smoothed models when we employ pre-training or semi-supervision. The results are summarized in Table 2.

## 4.3 Attacking trained models with SMOOTHADV

In this section, we assess the performance of our attack, particularly SMOOTHADV$_{\text{PGD}}$, for finding adversarial examples for the CIFAR-10 randomized smoothing models of Cohen et al. [6].

SMOOTHADV$_{\text{PGD}}$ requires the evaluation of (6) as discussed in Section 3. Here, we analyze how sensitive our attack is to the number of samples $m_{\text{test}}$ used in (6) for estimating the gradient of the adversarial objective. Fig. 3 shows the empirical accuracies for various values of $m_{\text{test}}$. Lower accuracies corresponds to stronger attack. SMOOTHADV with $m_{\text{test}} = 1$ sample performs worse than the vanilla PGD attack on the base classifier, but as $m_{\text{test}}$ increases, our attack becomes stronger, decreasing the gap between certified and empirical accuracies. We did not observe any noticeable improvement beyond $m_{\text{test}} = 128$. More details are in Appendix C.4.

While as discussed here, the success rate of the attack is affected by the number of Gaussian noise samples $m_{\text{test}}$ used by the attacker, it is also affected by the number of Gaussian noise samples $n$ in PREDICT used by the classifier. Indeed, as $n$ increases, abstention due to low confidence becomes more rare, increasing the prediction quality of the smoothed classifier. See a detailed analysis in Appendix C.5.

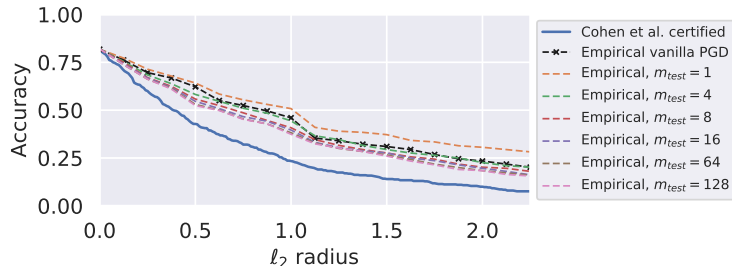

Figure 3: Certified and empirical robust accuracy of Cohen et al. [6]'s models on CIFAR-10. For each $\ell_2$ radius $r$, the certified/empirical accuracy is the maximum over randomized smoothing models trained using $\sigma \in \{0.12, 0.25, 0.5, 1.0\}$. The empirical accuracies are found using 20 steps of SMOOTHADV$_{\text{PGD}}$. The closer an empirical curve is to the certified curve, the stronger the corresponding attack is (the lower the better).

## 5    Related Work

Recently, many approaches (defenses) have been proposed to build adversarially robust classifiers, and these approaches can be broadly divided into *empirical* defenses and *certified* defenses.

**Empirical defenses** are empirically robust to existing adversarial attacks, and the best empirical defense so far is *adversarial training* [20, 25]. In this kind of defense, a neural network is trained to minimize the worst-case loss over a neighborhood around the input. Although such defenses seem powerful, nothing guarantees that a more powerful, not yet known, attack would not break them; the most that can be said is that known attacks are unable to find adversarial examples around the data points. In fact, most empirical defenses proposed in the literature were later "broken" by stronger adversaries [4, 2, 35, 1]. To stop this arms race between defenders and attackers, a number of work tried to focus on building certified defenses which enjoy formal robustness guarantees.

**Certified defenses** are provably robust to a specific class of adversarial perturbation, and can guarantee that for any input $x$, the classifier's prediction is constant within a neighborhood of $x$. These are typically based on certification methods which are either *exact* (a.k.a "complete") or *conservative* (a.k.a "sound but incomplete"). Exact methods, usually based on Satisfiability Modulo Theories solvers [18, 11] or mixed integer linear programming [34, 24, 12], are guaranteed to find an adversarial example around a datapoint if it exists. Unfortunately, they are computationally inefficient and difficult to scale up to large neural networks. Conservative methods are also guaranteed to detect an adversarial example if exists, but they might mistakenly flag a safe data point as vulnerable to adversarial examples. On the bright side, these methods are more scalable and efficient which makes some of them useful for building certified defenses [39, 36, 37, 27, 28, 40, 10, 9, 7, 30, 13, 26, 31, 15, 38, 41]. However, none of them have yet been shown to scale to practical networks that are large and expressive enough to perform well on ImageNet, for example. To scale up to practical networks, randomized smoothing has been proposed as a *probabilistically* certified defense.

**Randomized smoothing**   A randomized smoothing classifier is not itself a neural network, but uses a neural network as its base for classification. Randomized smoothing was proposed by several works [23, 3] as a heuristic defense without proving any guarantees. Lecuyer et al. [21] first proved robustness guarantees for randomized smoothing classifier, utilizing inequalities from the differential privacy literature. Subsequently, Li et al. [22] gave a stronger robustness guarantee using tools from information theory. Recently, Cohen et al. [6] provided a tight robustness guarantee for randomized smoothing and consequently achieved the state of the art in $\ell_2$-norm certified defense.

## 6    Conclusions

In this paper, we designed an adapted attack for smoothed classifiers, and we showed how this attack can be used in an adversarial training setting to substantially improve the provable robustness of smoothed classifiers. We demonstrated through extensive experimentation that our adversarially trained smooth classifiers consistently outperforms all existing provably $\ell_2$-robust classifiers by a significant margin on ImageNet and CIFAR-10, establishing the state of the art for provable $\ell_2$-defenses.

## Acknowledgements

We would like to thank Zico Kolter, Jeremy Cohen, Elan Rosenfeld, Aleksander Madry, Andrew Ilyas, Dimitris Tsipras, Shibani Santurkar, and Jacob Steinhardt for comments and discussions.

## Footnotes

[3]Note that we do not provide a new certification method incorporating adversarial training; the improvements that we get are due to the higher quality of our base classifiers as a result of adversarial training.

[4]Note that we are reusing the same noise samples during every step of our attack as well as during augmentation. Intuitively, this helps to stabilize the attack process.

[5]Similar to Cohen et al. [6], we certified the full CIFAR-10 test set and a subsampled ImageNet test set of 500 samples.

[6]We report the model with the highest certified $\ell_2$ accuracy on CIFAR-10 at a radius of 0.435, amongst all our models trained in this paper.

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
