[Supplementary Material]

# A Alternative proof of the robustness guarantee of Cohen et al. [6] via explicit Lipschitz constants of smoothed classifier

In this appendix, we present an alternate derivation of (2). Fix $f : \mathbb{R}^n \to [0, 1]$ and define $\hat{f}$ by:

$$\hat{f}(x) = (f * \mathcal{N}(0, I))(x) = \frac{1}{(2\pi)^{n/2}} \int_{\mathbb{R}^n} f(t) \exp\left(-\frac{1}{2}\|x - t\|^2\right) dt.$$

The smoothed function $\hat{f}$ is known as the *Weierstrass transform* of $f$, and a classical property of the Weierstrass transform is its induced smoothness, as demonstrated by the following.

**Lemma 1.** *The function $\hat{f}$ is $\sqrt{\frac{2}{\pi}}$-Lipschitz.*

*Proof.* It suffices to prove that for any unit direction $u$ one has $u \cdot \nabla \hat{f}(x) \le \sqrt{\frac{2}{\pi}}$. Note that:

$$\nabla \hat{f}(x) = \frac{1}{(2\pi)^{n/2}} \int_{\mathbb{R}^n} f(t)(x - t) \exp\left(-\frac{1}{2}\|x - t\|^2\right) dt, \tag{8}$$

and thus (using $|f(t)| \le 1$, and classical integration of the Gaussian density)

$$\begin{aligned}
u \cdot \nabla \hat{f}(x) &\le \frac{1}{(2\pi)^{n/2}} \int_{\mathbb{R}^n} |u \cdot (x - t)| \exp\left(-\frac{1}{2}\|x - t\|^2\right) dt \\
&= \frac{1}{\sqrt{2\pi}} \int_{-\infty}^{+\infty} |s| \exp\left(-\frac{1}{2}s^2\right) ds = \sqrt{\frac{2}{\pi}}.
\end{aligned}$$

$\square$

However, $\hat{f}$ in fact satisfies an even stronger *nonlinear* smoothness property as shown in the following lemma.

**Lemma 2.** *Let $\Phi(a) = \frac{1}{\sqrt{2\pi}} \int_{-\infty}^{a} \exp\left(-\frac{1}{2}s^2\right) ds$. For any function $f : \mathbb{R}^n \to [0, 1]$, the map $x \mapsto \Phi^{-1}(\hat{f}(x))$ is 1-Lipschitz.*

*Proof.* Note that:

$$\nabla \Phi^{-1}(\hat{f}(x)) = \frac{\nabla \hat{f}(x)}{\Phi'(\Phi^{-1}(\hat{f}(x)))},$$

and thus we need to prove that for any unit direction $u$, denoting $p = \hat{f}(x)$,

$$u \cdot \nabla \hat{f}(x) \le \frac{1}{\sqrt{2\pi}} \exp\left(-\frac{1}{2}(\Phi^{-1}(p))^2\right).$$

Note that the left-hand side can be written as follows (recall (8))

$$\mathbb{E}_{X \sim \mathcal{N}(0, I_n)}[f(x + X)X \cdot u].$$

We now claim that the supremum of the above quantity over all functions $f : \mathbb{R}^n \to [0, 1]$, subject to the constraint that $\mathbb{E}[f(x + X)] = p$, is equal to:

$$\mathbb{E}[(X \cdot u)\mathbb{1}\{X \cdot u \ge -\Phi^{-1}(p)\}] = \frac{1}{\sqrt{2\pi}} \exp\left(-\frac{1}{2}(\Phi^{-1}(p)^2)\right), \tag{9}$$

which would conclude the proof.

To see why the latter claim is true, first notice that $h : x \mapsto \mathbb{1}\{x \cdot u \ge -\Phi^{-1}(p)\}$ achieves equality. Let us assume by contradiction that the maximizer is obtained at some function $f : \mathbb{R}^n \to [0, 1]$ different from $h$. Consider the set $\Omega^+$ where $h(x) > f(x)$ and $\Omega^-$ the set where $h(x) < f(x)$, and note that since both functions integrate to $p$, it must be that $\int_{\Omega^+} (h - f)d\mu = \int_{\Omega^-} (f - h)d\mu$ (where $\mu$ is the Gaussian measure). Now simply consider the new function $\tilde{f} = f + (h - f)\mathbb{1}\{\Omega^+\} - (f - h)\mathbb{1}\{\Omega^-\}$. Note that $\tilde{f}$ takes value in $[0, 1]$ and integrates to $p$. Moreover, denoting $g(x) = x \cdot u$, one has $\int fg d\mu < \int \tilde{f} g d\mu$. Indeed, by definition of $h$, one has for any $x \in \Omega_+$ and $y \in \Omega^-$ that $g(x) > g(y)$. This concludes the proof. $\square$

It turns out that the smoothness property of lemma 2 naturally leads to the robustness guarantee (2) of Cohen et al. [6]. To see why, let $\hat{f}_i : \mathbb{R}^n \to [0, 1]$ be the output of the smoothed classifier mapping a point $x \in \mathbb{R}^n$ to the probability of it belonging to class $c_i$. Assume that the smooth classifier assigns to $x$ the class $c_A$ with probability $p_A = \hat{f}_A(x)$. Denote by $c_B$ any other class such that $c_B \neq c_A$ and $p_B = \hat{f}_B(x) \leq p_A$. By lemma 2, we know that under any perturbation $\delta \in \mathbb{R}^n$ of $x$,

$$\Phi^{-1}\left(\hat{f}_A(x)\right) - \Phi^{-1}\left(\hat{f}_A(x + \delta)\right) \leq \|\delta\|_2. \tag{10}$$

For an adversarial $\delta$, $\hat{f}_A(x + \delta) \leq \hat{f}_B(x + \delta)$ for some class $c_B$, leading to

$$\Phi^{-1}\left(\hat{f}_A(x)\right) - \Phi^{-1}\left(\hat{f}_B(x + \delta)\right) \leq \|\delta\|_2. \tag{11}$$

By lemma 2 applied to $\hat{f}_B$, and noting that $\hat{f}_B(x + \delta) \geq \hat{f}_B(x)$ , we know that,

$$\Phi^{-1}\left(\hat{f}_B(x + \delta)\right) - \Phi^{-1}\left(\hat{f}_B(x)\right) \leq \|\delta\|_2. \tag{12}$$

Combining (11) and (12), it is straightforward to see that

$$\|\delta\|_2 \geq \frac{1}{2}\left(\Phi^{-1}(p_A) - \Phi^{-1}(p_B)\right) \tag{13}$$

The above equation gives a lower bound on the minimum $\ell_2$ adversarial perturbation required to flip the classification from $c_A$ to $c_B$. This lower bound is minimized when $p_B$ is maximized over the set of classes $C \setminus \{c_A\}$. Therefore, $c_B$ is the runner up class returned by the smoothed classifier at $x$. Finally, the factor $\sigma$ that appears in (2) can be obtained by re-deriving the above with $\hat{f}(x) = \left(f * \mathcal{N}(0, \sigma^2 I)\right)(x)$ and $\Phi(a) = \frac{1}{\sqrt{2\pi}}\int_{-\infty}^{a} \exp\left(-\frac{1}{2}(\frac{s}{\sigma})^2\right) ds$.

Note that both lemmas presented in this appendix give the same robustness guarantee for small gaps $(p_A - p_B)$, but the second lemma is much better for large gaps (in fact, in the limit of a gap going to 1, the second lemma gives an infinite radius while the first lemma only gives a radius of $\frac{1}{2}\sqrt{\frac{\pi}{2}}$).

## B    Another perspective for deriving SMOOTHADV

In this section we provide an alternative motivation for the SMOOTHADV objective presented in Section 2.2. We assume that we have a hard classifier $f : \mathbb{R}^d \to \mathcal{Y}$ which takes the form $f(x) = \arg\max_{y \in \mathcal{Y}} L(x)_y$, for some function $L : \mathbb{R}^d \to \mathbb{R}^{\mathcal{Y}}$. If $f$ is a neural network classifier, this $L$ can be taken for instance to be the map from the input to the logit layer immediately preceding the softmax. If $f$ is of this form, then the smoothed soft classifier $g$ with parameter $\sigma^2$ associated to (the one-hot encoding of) $f$ can be written has

$$
\begin{aligned}
g(x)_y &= \Pr_{\delta \sim \mathcal{N}(0,\sigma^2 I)}\left[\arg\max_{y' \in \mathcal{Y}} L(x + \delta)_{y'} = y\right] \\
&= \mathbb{E}_{\delta \sim \mathcal{N}(0,\sigma^2 I)}\left[\nu(L(x + \delta))_y\right] ,
\end{aligned}
\tag{14}
$$

for all $y \in \mathcal{Y}$, where $\nu : \mathbb{R}^d \to \mathbb{R}^{\mathcal{Y}}$ is the function, which at input $z$, has $y$-th coordinate equal to 1 if and only if $y = \arg\max_{y' \in \mathcal{Y}} z_{y'}$, and zero otherwise. The function $\nu$ is somewhat hard to work with, therefore we will approximate it with a smooth function, namely, the softmax function. Recall that the softmax function with inverse temperature parameter $\beta$ is the function $\zeta_\beta : \mathbb{R}^{\mathcal{Y}} \to P(\mathcal{Y})$ given by $\zeta_\beta(z)_y = e^{\beta z_y} / \sum_{y' \in \mathcal{Y}} e^{\beta z_{y'}}$. Observe that for any $z \in \mathbb{R}^{\mathcal{Y}}$, we have that $\zeta_\beta(z) \to \nu(z)$ as $\beta \to \infty$. Thus we can approximate (14) with

$$g(x)_y \approx \mathbb{E}_{\delta \sim \mathcal{N}(0,\sigma^2 I)}\left[\zeta_\beta(L(x + \delta))_y\right] . \tag{15}$$

To find an adversarial perturbation of $g$ at data point $(x, y)$, it is sufficient to find a perturbation $\hat{x}$ so that $g(x)_y$ is minimized. Combining this with the approximation (15), we find that a heuristic to find an adversarial example for the smoothed classifier at $(x, y)$ is to solve the following optimization problem:

$$\hat{x} = \arg\min_{\|x'-x\|_2 \leq \epsilon} \mathbb{E}_{\delta \sim \mathcal{N}(0,\sigma^2 I)}\left[\zeta_\beta(L(x' + \delta))_y\right] , \tag{16}$$

and as we let $\beta \rightarrow \infty$, this converges to finding an adversarial example for the true smoothed classifier.

To conclude, we simply observe that for neural networks, $\zeta_\beta(L(x+\delta))_y$ is exactly the soft classifier that is thresholded to form the hard classifier, if $\beta$ is taken to be 1. Therefore the solution to $(\mathcal{S})$ and (16) with $\beta = 1$ are the same, since log is a monotonic function.

An interesting direction is to investigate whether varying $\beta$ in (16) allows us to improve our adversarial attacks, and if they do, whether this gives us stronger adversarial training as well. Intuitively, as we take $\beta \rightarrow \infty$, the quality of the optimal solution should increase, but the optimization problem becomes increasingly ill-behaved, and so it is not clear if the actual solution we obtain to this problem via first order methods becomes better or not.

## C  Additional Experiments

### C.1  Adversarial attacking the base model instead of the smoothed model

We compare SMOOTHADV-ersarial training (training the smoothed classifier $g$) to:

1. using vanilla adversarial training (PGD) to find adversarial examples of the *base classifier* $f$ and train on them. We refer to this as **Vanilla PGD** training.

2. using vanilla adversarial training (PGD) to find adversarial examples of the *base classifier* $f$, add Gaussian noise to them, then train on the resulting inputs. We refer to this as **Vanilla PGD+noise** training.

For our method and the above two methods, we use $T = 2$ steps of attack, $m_{train} = 1$, and we train for $\epsilon \in \{0.25, 0.5, 1.0, 2.0\}$, and for $\sigma \in \{0.12, 0.25, 0.5, 1.0\}$.

Fig. 4 plots the best certified accuracies over all $\epsilon$ and $\sigma$ values, for each $\ell_2$ radius $r$ using our SMOOTHADV$_{\text{PGD}}$ trained classifiers vs. smoothed models trained via Vanilla PGD or Vanilla PGD+noise. Fig. 4 also plots Cohen et al. [6] results as a baseline. Observe that SMOOTHADV-ersially trained models are more robust overall.

Figure 4: Certified defenses: ours vs. Cohen et al. [6] vs. vanilla PGD vs. vanilla PGD + noise.

### C.2  Effect of number of noise samples $m_{train}$ in (6) during SMOOTHADV-ersarial training on the certified accuracy of smoothed classifiers

As presented in Section 4.3, more noise samples $\delta_i$ lead to stronger SMOOTHADV-eraial attack. Here, we demonstrate that if we train with such improved attacks, we get higher certified accuracies of the smoothed classifier. Fig. 5 plots the best certified accuracies over models trained using SMOOTHADV$_{\text{PGD}}$ or SMOOTHADV$_{\text{DDN}}$ with $T \in \{2, 4, 6, 8, 10\}$, $\sigma \in \{0.12, 0.25, 0.5, 1.0\}$, $\epsilon \in \{0.25, 0.5, 1.0, 2.0\}$, and across various number of noise samples $m_{train}$ for the attack. Observe that models trained with higher $m_{train}$ tend to have higher certified accuracies.

Figure 5: Vary number of samples $m_{train}$.

## C.3    Effect of $\epsilon$ during training on the certified accuracy of smoothed classifiers

Here, we analyze the effect of the maximum allowed $\ell_2$ perturbation of SMOOTHADV during adversarial training on the robustness of the obtained smoothed classifier. Fig. 6 plots the best certified accuracies for $\epsilon \in \{0.25, 0.5, 1.0, 2.0\}$ over models trained using SMOOTHADV$_{PGD}$ with $T \in \{2, 4, 6, 8, 10\}$, $m_{train} \in \{1, 2, 4, 8\}$, and $\sigma \in \{0.12, 0.25, 0.5, 1.0\}$. Observe that as $\epsilon$ increases, the certified accuracies for small $\ell_2$ radii decrease, but those for large $\ell_2$ radii increase, which is expected.

Figure 6: Vary $\epsilon$. Observe that as $\epsilon$ increases, the certified accuracies for small $\ell_2$ radii decrease, but those for large $\ell_2$ radii increase, which is expected.

## C.4    Effect of the number of samples $m_{test}$ in (6) during SMOOTHADV attack on the empirical accuracies

SMOOTHADV$_{PGD}$ requires the evaluation of (6) as discussed in Section 3. Here, we analyze how sensitive our attack is to the number of samples $m_{test}$ used in (6). Fig. 7 shows the empirical accuracies for various values of $m_{test}$. Lower accuracies correspond to stronger attacks. For $m_{test} = 1$, the vanilla PGD attack (attacking the base classifier instead of the smooth classifier) performs better than SMOOTHADV, but as $m_{test}$ increases, our attack becomes stronger, decreasing the gap between certified and empirical accuracies. We did not observe any noticeable improvement beyond $m_{test} = 128$.

Figure 7: **(A larger version of Fig. 3)** Certified and empirical robust accuracy of Cohen et al. [6]'s models on CIFAR-10. For each $\ell_2$ radius $r$, the certified/empirical accuracy is the maximum over randomized smoothing models trained using $\sigma \in \{0.12, 0.25, 0.5, 1.0\}$. The empirical accuracies are found using 20 steps of SMOOTHADV$_{PGD}$. The closer an empirical curve is to the certified curve, the stronger the corresponding attack is (the lower the better).

## C.5 Effect of the number of Monte Carlo samples $n$ in PREDICT on the empirical accuracies

Fig. 8 plots the empirical accuracies of $g$ using a SMOOTHADV$_{PGD}$ attack (with $m_{test} = 128$) across different numbers of Monte Carlo samples n that are used by PREDICT. Observe that the empirical accuracies increase as $n$ increases since the prediction quality of the smoothed classifier improves i.e. less predictions are abstained.

Figure 8: Empirical accuracies. Vary number of samples $n$. The higher the better.

## C.6 Performance of the gradient-free estimator (7)

Despite the appealing features of the gradient-free estimator (7) presented in Section 3.1 as an alternative to (6), in practice we find that this attack is quite weak. This is shown in Fig. 9 for various values of $m_{test}$.

We speculate that this is because the variance of the gradient estimator is too high. We believe that investigating this attack in practice is an interesting direction for future work.

Figure 9: The emprirical accuracies found by the attack ($\mathcal{S}$) using the plug-in estimator (6) vs. the gradient-free estimator (7). The closer an empirical curve is to the certified curve, the stronger the attack.

Table 4: The certification abstention rate of our best CIFAR-10 classifiers at various $\ell_2$ radii.

| $\ell_2$ RADIUS (CIFAR-10) | 0.25 | 0.5 | 0.75 | 1.0 | 1.25 | 1.5 | 1.75 | 2.0 | 2.25 |
|---|---|---|---|---|---|---|---|---|---|
| COHEN ET AL. [6] (%) | 8.0 | 15.2 | 15.2 | 29.5 | 29.5 | 29.5 | 29.5 | 29.5 | 29.5 |
| OURS (%) | 1.7 | 3.3 | 3.0 | 5.8 | 4.7 | 4.7 | 4.3 | 13.2 | 13.2 |
| + PRE-TRAINING (%) | 0.9 | 2.9 | 2.1 | 3.9 | 3.9 | 3.9 | 3.9 | 10.4 | 10.4 |
| + SEMI-SUPERVISION (%) | 1.1 | 2.8 | 2.8 | 6.3 | 6.3 | 4.4 | 4.3 | 11.5 | 11.5 |
| + BOTH(%) | 1.0 | 2.7 | 2.1 | 4.1 | 4.1 | 4.1 | 4.1 | 11.6 | 11.6 |

## C.7 Certification Abstention Rate

In this section, we compare the certification abstention rates of our smoothed models against those of Cohen et al. [6]'s models. Table 4 reports the abstention rates for the best models at various $\ell_2$-radii. These are the models corresponding to Table 2. Our models have a substantially lower abstention rate across all $\ell_2$ radii.

Note that Cohen et al. [6] reported the abstention rates for *prediction* (but not certification), which tend to be lower than the certification abstention rates.

## D  Experiments Details

Here we include details of all the experiments conducted in this paper.

**Attacks used in the paper**  We use two of the strongest attacks in the literature, projected gradient descent (PGD) [25] and decoupled direction and norm (DDN) [29] attacks. We adapt these attacks such that their gradient steps are given by (6), and we call the resulting attacks SMOOTHADV$_{\text{PGD}}$ and SMOOTHADV$_{\text{DDN}}$, respectively.

For PGD (SMOOTHADV$_{\text{PGD}}$), we use a constant step size $\gamma = 2\frac{\epsilon}{T}$ where $T$ is the number of attack steps, and $\epsilon$ is the maximum allowed $\ell_2$ perturbation of the input.

For DDN (SMOOTHADV$_{\text{DDN}}$), the attack objective is in fact different than that of PGD (i.e. different that ($\mathcal{S}$)). DDN tries to find the "closest" adversarial example to the input instead of finding the "best" adversarial example (in terms of maximizing the loss in a given neighborhood of the input). We stick to the hyperparameters used in the original paper [29]. We use $\epsilon_0 = 1$, $\gamma = 0.05$, and an initial step size $\alpha = 1$ that is reduced with cosine annealing to 0.01 in the last iteration (see [29] for the definition of these parameters). We experimented with very few iterations ($\{2, 4, 6, 8, 10\}$) as compared to the original paper, but we still got good results.

We emphasize that we are not using PGD and DDN to attack the *base classifer* $f$ of a smoothed model, instead we are using them to adversarially train *smoothed classiers* (see Pseudocode 1).

**Training details** In order to report certified radii in the original coordinates, we first added Gaussian noise and/or do adversarial attacks, and then standardized the data (in contrast to importing a standardized dataset). Specifically, in our PyTorch implementation, the first layer of the base classifier is a normalization layer that performed a channel-wise standardization of its input.

For both ImageNet and CIFAR-10, we trained the base classifier with random horizontal flips and random crops (in addition to the Gaussian data augmentation discussed in Section 3.2).

The main training algorithm is shown in Pseudocode 1. It has the following parameters: $B$ is the mini-batch size, $m$ is the number of noise samples used for gradient estimation in (6) as well as for Gaussian noise data augmentation, and $T$ is the number of steps of an attack.

We point out few remarks.

1. First, an important parameter is the radius of the attack $\epsilon$. During the first epoch, it is set to zero, then we linearly increase it over the first ten epochs, then it stays constant.

2. Second, we are reusing the same noise samples during every step of our attack as well as augmentation. Intuitively, it helps to stabilize the attack process.

3. Finally, the way training is described in Pseudocode 1 is not efficient; it needs to be appropriately batched so that we compute adversarial examples for every input in a batch at the same time.

**Compute details and training time** On CIFAR-10, we trained using SGD on one NVIDIA P100 GPU. We train for 150 epochs. We use a batch size of 256, and an initial learning rate of 0.1 which drops by a factor of 10 every 50 epochs. Training time varies between few hours to few days, depending on how many attack steps $T$ and noise samples $m$ are used in Pseudocode 1.

On ImageNet we trained with synchronous SGD on four NVIDIA V100 GPUs. We train for 90 epochs. We use a batch size of 400, and an initial learning rate of 0.1 which drops by a factor of 10 every 30 epochs. Training time varies between 2 to 6 days depending on whether we are doing SMOOTHADV-ersarial training or just Gaussian noise training (similar to Cohen et al. [6]).

**Models used** The models used in this paper are similar to those used in Cohen et al. [6]: a ResNet-50 [16] on ImageNet, and ResNet-110 on CIFAR-10. These models can be found on the github repo accompanying [6] `https://github.com/locuslab/smoothing/blob/master/code/architectures.py`.

**Parameters of CERTIFY amd PREDICT** For details of these algorithms, please see the *Pseudocode* in [6].

For CERTIFY, unless otherwise specified, we use $n = 100,000$, $n_0 = 100$, $\alpha = 0.001$.

For PREDICT, unless otherwise specified, we use $n = 100,000$ and $\alpha = 0.001$.

**Source code** Our code and trained models are publicly available at `http://github.com/Hadisalman/smoothing-adversarial`. The repository also includes all our training/certification logs, which enables the replication of all the results of this paper by running a single piece of code. Check the repository for more details.

# E Details for Pre-training and Semi-supervision to Improve the Provable Robustness

## E.1 Pre-training

In this appendix, we describe the details of how we employ pre-training within our framework to boost the certified robustness of our models. We pretrain smoothed classifiers on a 32x32 down-sampled version of ImageNet (ImageNet32) as done by Hendrycks et al. [17]. Then we fine-tune all the weights of these models on CIFAR-10 (with the 1000-dimensional logit layer of each model replaced by a randomly initialized 10-dimensional logit layer suitable for CIFAR-10).

**ImageNet32 training** We train ResNet-110 architectures on ImageNet32 using SGD on one NVIDIA P100 GPU. We train for 150 epochs. We use a batch size of 256, and an initial learn-

ing rate of 0.1 which drops by a factor of 10 every 50 epochs. We use SMOOTHADV$_\text{PGD}$ with $T = 2$ steps and $m_{train} = 1$ noise samples. We train a total of 16 models each corresponding to a choice of $\sigma \in \{0.12, 0.25, 0.5, 1.0\}$ and $\epsilon \in \{0.25, 0.5, 1.0, 2.0\}$.

**Fine-tuning on CIFAR-10**  For each choice of $\sigma$ and $\epsilon$, we fine tune the corresponding ImageNet32 model on CIFAR-10; we replace the 1000-dimensional logit layer of each model with a randomly initialized 10-dimensional logit layer suitable for CIFAR-10, then we train for 30 epochs with a constant learning rate of 0.001 and a batch size of 256. We use SMOOTHADV$_\text{PGD}$ with $T \in \{2, 4, 6, 8, 10\}$ and $m_{train} \in \{1, 2, 4, 8\}$.

### E.2  Semi-supervised Learning

In this appendix, we detail how we employ semi-supervised learning [5] within our framework to boost the certified robustness of our models.

We train our CIFAR-10 smoothed classifiers via the *self-training* technique of [5] using their 500K unlabelled dataset. We equip this dataset with pseudo-labels generated by a standard neural network trained on CIFAR-10, as in [5]; see [5] for more details[7].

Self-training a smoothed classifier works as follows: at every step we randomly sample either a labelled minibatch from CIFAR-10, or a pseudo-labelled minibatch from the 500K dataset:

1. for a labelled minibatch, we follow Pseudocode 1 as is.
2. for a pseudo-labelled minibatch, we scale the CE loss by a factor of $\eta \in \{0.1, 0.5, 1.0\}$ and we follow the rest of Pseudocode 1.

We use SMOOTHADV$_\text{PGD}$ with $T \in \{2, 4, 6, 8, 10\}$, $m_{train} = 1$, $\sigma \in \{0.12, 0.25, 0.5, 1.0\}$, and $\epsilon \in \{0.25, 0.5, 1.0, 2.0\}$.

### E.3  Semi-supervised Learning with Pre-training

We also experiment with combining semi-supervised learning with pre-training in the hopes of obtaining further improvements. We start from the same ResNet-110 models pretrained on ImageNet32 as in Appendix E.1. Then we finetune these models using semi-supervision, as in Appendix E.2, for 30 epochs with a learning rate of 0.001. We use SMOOTHADV$_\text{PGD}$ with $T \in \{2, 4, 6, 8, 10\}$, $m_{train} = 1$, $\sigma \in \{0.12, 0.25, 0.5, 1.0\}$, and $\epsilon \in \{0.25, 0.5, 1.0, 2.0\}$.

## F  $\ell_2$ to $\ell_\infty$ Certified Defense on ImageNet

We find our $\ell_2$-robust ImageNet models enjoy non-trivial $\ell_\infty$ certified robustness. In Table 5, we report the best $\ell_\infty$ certified accuracy that we get at a radius of 1/255 (implied by the $\ell_2$ certified accuracy at a radius of $1.5 \approx \sqrt{3 \times 224^2}/255$). We exceed previous state-of-the-art in certified $\ell_\infty$ defenses by around $8.2\%$.

Table 5: Certified $\ell_\infty$ robustness at a radius of $\frac{1}{255}$ on ImageNet.

| MODEL | $\ell_\infty$ ACC. AT 1/255 | STANDARD ACC. |
|---|---|---|
| OURS (%) | **38.2** | 54.6 |
| COHEN ET AL. [6] (%) | 28.6 | **57.2** |

# G    ImageNet and CIFAR-10 Detailed Results

In this appendix, we include the certified accuracies of each mode that we use in the paper. For each $\ell_2$ radius, we highlight the best accuracy across all models. Note that we outperform the models of Cohen et al. [6] (first three rows of each table) over all $\ell_2$ radii by wide margins.

Table 6: Approximate certified test accuracy on ImageNet. Each row is a setting of the hyperparameters $\sigma$ and $\epsilon$, each column is an $\ell_2$ radius. The entry of the best $\sigma$ for each radius is bolded. For comparison, random guessing would attain 0.001 accuracy.

| | $\ell_2$ RADIUS (IMAGENET) | | 0.0 | 0.5 | 1.0 | 1.5 | 2.0 | 2.5 | 3.0 | 3.5 |
|---|---|---|---|---|---|---|---|---|---|---|
| [6] | $\sigma = 0.25$ | | **0.67** | 0.49 | 0.00 | 0.00 | 0.00 | 0.00 | 0.00 | 0.00 |
| | $\sigma = 0.50$ | | 0.57 | 0.46 | 0.37 | 0.29 | 0.00 | 0.00 | 0.00 | 0.00 |
| | $\sigma = 1.00$ | | 0.44 | 0.38 | 0.33 | 0.26 | 0.19 | 0.15 | 0.12 | 0.09 |
| SMOOTHADV$_{\text{PGD}}$ | $\sigma = 0.25$ | $\epsilon = 0.5$ | 0.63 | 0.54 | 0.00 | 0.00 | 0.00 | 0.00 | 0.00 | 0.00 |
| | $\sigma = 0.25$ | $\epsilon = 1.0$ | 0.62 | 0.54 | 0.00 | 0.00 | 0.00 | 0.00 | 0.00 | 0.00 |
| | $\sigma = 0.25$ | $\epsilon = 2.0$ | 0.56 | 0.52 | 0.00 | 0.00 | 0.00 | 0.00 | 0.00 | 0.00 |
| | $\sigma = 0.25$ | $\epsilon = 4.0$ | 0.49 | 0.45 | 0.00 | 0.00 | 0.00 | 0.00 | 0.00 | 0.00 |
| | $\sigma = 0.50$ | $\epsilon = 0.5$ | 0.56 | 0.48 | 0.42 | 0.34 | 0.00 | 0.00 | 0.00 | 0.00 |
| | $\sigma = 0.50$ | $\epsilon = 1.0$ | 0.54 | 0.49 | **0.43** | **0.37** | 0.00 | 0.00 | 0.00 | 0.00 |
| | $\sigma = 0.50$ | $\epsilon = 2.0$ | 0.48 | 0.45 | 0.42 | 0.37 | 0.00 | 0.00 | 0.00 | 0.00 |
| | $\sigma = 0.50$ | $\epsilon = 4.0$ | 0.44 | 0.42 | 0.39 | 0.37 | 0.00 | 0.00 | 0.00 | 0.00 |
| | $\sigma = 1.00$ | $\epsilon = 0.5$ | 0.44 | 0.38 | 0.34 | 0.29 | 0.24 | 0.20 | 0.15 | 0.11 |
| | $\sigma = 1.00$ | $\epsilon = 1.0$ | 0.41 | 0.36 | 0.34 | 0.31 | 0.26 | 0.21 | 0.18 | 0.14 |
| | $\sigma = 1.00$ | $\epsilon = 2.0$ | 0.40 | 0.37 | 0.34 | 0.30 | **0.27** | **0.25** | **0.20** | 0.15 |
| | $\sigma = 1.00$ | $\epsilon = 4.0$ | 0.34 | 0.31 | 0.29 | 0.27 | 0.25 | 0.22 | 0.19 | **0.16** |
| SMOOTHADV$_{\text{DDN}}$ | $\sigma = 0.25$ | $\epsilon = 0.5$ | 0.66 | 0.52 | 0.00 | 0.00 | 0.00 | 0.00 | 0.00 | 0.00 |
| | $\sigma = 0.25$ | $\epsilon = 1.0$ | 0.65 | **0.56** | 0.00 | 0.00 | 0.00 | 0.00 | 0.00 | 0.00 |
| | $\sigma = 0.25$ | $\epsilon = 2.0$ | 0.65 | 0.54 | 0.00 | 0.00 | 0.00 | 0.00 | 0.00 | 0.00 |
| | $\sigma = 0.25$ | $\epsilon = 4.0$ | **0.67** | 0.55 | 0.00 | 0.00 | 0.00 | 0.00 | 0.00 | 0.00 |
| | $\sigma = 0.50$ | $\epsilon = 0.5$ | 0.59 | 0.48 | 0.38 | 0.29 | 0.00 | 0.00 | 0.00 | 0.00 |
| | $\sigma = 0.50$ | $\epsilon = 1.0$ | 0.55 | 0.49 | 0.40 | 0.32 | 0.00 | 0.00 | 0.00 | 0.00 |
| | $\sigma = 0.50$ | $\epsilon = 2.0$ | 0.58 | 0.49 | 0.42 | 0.34 | 0.00 | 0.00 | 0.00 | 0.00 |
| | $\sigma = 0.50$ | $\epsilon = 4.0$ | 0.58 | 0.51 | 0.41 | 0.32 | 0.00 | 0.00 | 0.00 | 0.00 |
| | $\sigma = 1.00$ | $\epsilon = 0.5$ | 0.44 | 0.37 | 0.31 | 0.26 | 0.20 | 0.16 | 0.11 | 0.08 |
| | $\sigma = 1.00$ | $\epsilon = 1.0$ | 0.46 | 0.39 | 0.32 | 0.26 | 0.22 | 0.17 | 0.11 | 0.09 |
| | $\sigma = 1.00$ | $\epsilon = 2.0$ | 0.45 | 0.39 | 0.34 | 0.27 | 0.23 | 0.16 | 0.13 | 0.09 |
| | $\sigma = 1.00$ | $\epsilon = 4.0$ | 0.44 | 0.39 | 0.34 | 0.28 | 0.22 | 0.16 | 0.12 | 0.08 |

Table 7: SMOOTHADV-ersarial training $T = 2$ steps, $m_{train} = 1$ sample.

| | $\ell_2$ RADIUS (CIFAR-10) | | 0.0 | 0.25 | 0.5 | 0.75 | 1.0 | 1.25 | 1.5 | 1.75 | 2.0 | 2.25 |
|---|---|---|---|---|---|---|---|---|---|---|---|---|
| [6] | $\sigma = 0.12$ | | 0.81 | 0.59 | 0.00 | 0.00 | 0.00 | 0.00 | 0.00 | 0.00 | 0.00 | 0.00 |
| | $\sigma = 0.25$ | | 0.75 | 0.60 | 0.43 | 0.27 | 0.00 | 0.00 | 0.00 | 0.00 | 0.00 | 0.00 |
| | $\sigma = 0.50$ | | 0.65 | 0.55 | 0.41 | 0.32 | 0.23 | 0.15 | 0.09 | 0.05 | 0.00 | 0.00 |
| | $\sigma = 1.00$ | | 0.47 | 0.39 | 0.34 | 0.28 | 0.22 | 0.17 | 0.14 | 0.12 | 0.10 | 0.08 |
| SMOOTHADV$_{\text{PGD}}$ | $\sigma = 0.12$ | $\epsilon = 0.25$ | **0.84** | **0.69** | 0.00 | 0.00 | 0.00 | 0.00 | 0.00 | 0.00 | 0.00 | 0.00 |
| | $\sigma = 0.12$ | $\epsilon = 0.50$ | 0.75 | 0.63 | 0.00 | 0.00 | 0.00 | 0.00 | 0.00 | 0.00 | 0.00 | 0.00 |
| | $\sigma = 0.12$ | $\epsilon = 1.00$ | 0.75 | 0.63 | 0.00 | 0.00 | 0.00 | 0.00 | 0.00 | 0.00 | 0.00 | 0.00 |
| | $\sigma = 0.12$ | $\epsilon = 2.00$ | 0.78 | 0.64 | 0.00 | 0.00 | 0.00 | 0.00 | 0.00 | 0.00 | 0.00 | 0.00 |
| | $\sigma = 0.25$ | $\epsilon = 0.25$ | 0.77 | 0.65 | 0.49 | 0.33 | 0.00 | 0.00 | 0.00 | 0.00 | 0.00 | 0.00 |
| | $\sigma = 0.25$ | $\epsilon = 0.50$ | 0.70 | 0.57 | 0.50 | 0.40 | 0.00 | 0.00 | 0.00 | 0.00 | 0.00 | 0.00 |
| | $\sigma = 0.25$ | $\epsilon = 1.00$ | 0.72 | 0.58 | 0.45 | 0.35 | 0.00 | 0.00 | 0.00 | 0.00 | 0.00 | 0.00 |
| | $\sigma = 0.25$ | $\epsilon = 2.00$ | 0.71 | 0.60 | 0.48 | 0.36 | 0.00 | 0.00 | 0.00 | 0.00 | 0.00 | 0.00 |
| | $\sigma = 0.50$ | $\epsilon = 0.25$ | 0.66 | 0.55 | 0.44 | 0.34 | 0.25 | 0.18 | 0.12 | 0.08 | 0.00 | 0.00 |
| | $\sigma = 0.50$ | $\epsilon = 0.50$ | 0.68 | 0.55 | 0.49 | 0.33 | 0.26 | 0.17 | 0.11 | 0.10 | 0.00 | 0.00 |
| | $\sigma = 0.50$ | $\epsilon = 1.00$ | 0.63 | 0.52 | 0.44 | 0.33 | 0.25 | 0.18 | 0.14 | 0.09 | 0.00 | 0.00 |
| | $\sigma = 0.50$ | $\epsilon = 2.00$ | 0.63 | 0.52 | 0.44 | 0.36 | 0.28 | 0.20 | 0.15 | 0.10 | 0.00 | 0.00 |
| | $\sigma = 1.00$ | $\epsilon = 0.25$ | 0.50 | 0.42 | 0.34 | 0.27 | 0.22 | 0.19 | 0.15 | 0.13 | 0.10 | 0.07 |
| | $\sigma = 1.00$ | $\epsilon = 0.50$ | 0.47 | 0.39 | 0.34 | 0.27 | 0.23 | 0.18 | 0.16 | 0.13 | 0.11 | 0.08 |
| | $\sigma = 1.00$ | $\epsilon = 1.00$ | 0.48 | 0.41 | 0.35 | 0.29 | 0.25 | 0.20 | 0.16 | 0.14 | 0.12 | 0.09 |
| | $\sigma = 1.00$ | $\epsilon = 2.00$ | 0.43 | 0.40 | 0.34 | 0.28 | 0.25 | 0.21 | 0.17 | 0.14 | 0.12 | 0.10 |
| SMOOTHADV$_{\text{DDN}}$ | $\sigma = 0.12$ | $\epsilon = 0.25$ | 0.82 | **0.69** | 0.00 | 0.00 | 0.00 | 0.00 | 0.00 | 0.00 | 0.00 | 0.00 |
| | $\sigma = 0.12$ | $\epsilon = 0.50$ | 0.79 | 0.67 | 0.00 | 0.00 | 0.00 | 0.00 | 0.00 | 0.00 | 0.00 | 0.00 |
| | $\sigma = 0.12$ | $\epsilon = 1.00$ | 0.71 | 0.64 | 0.00 | 0.00 | 0.00 | 0.00 | 0.00 | 0.00 | 0.00 | 0.00 |
| | $\sigma = 0.12$ | $\epsilon = 2.00$ | 0.54 | 0.49 | 0.00 | 0.00 | 0.00 | 0.00 | 0.00 | 0.00 | 0.00 | 0.00 |
| | $\sigma = 0.25$ | $\epsilon = 0.25$ | 0.77 | 0.65 | 0.51 | 0.39 | 0.00 | 0.00 | 0.00 | 0.00 | 0.00 | 0.00 |
| | $\sigma = 0.25$ | $\epsilon = 0.50$ | 0.74 | 0.64 | **0.53** | 0.41 | 0.00 | 0.00 | 0.00 | 0.00 | 0.00 | 0.00 |
| | $\sigma = 0.25$ | $\epsilon = 1.00$ | 0.64 | 0.59 | 0.53 | **0.45** | 0.00 | 0.00 | 0.00 | 0.00 | 0.00 | 0.00 |
| | $\sigma = 0.25$ | $\epsilon = 2.00$ | 0.53 | 0.49 | 0.46 | 0.42 | 0.00 | 0.00 | 0.00 | 0.00 | 0.00 | 0.00 |
| | $\sigma = 0.50$ | $\epsilon = 0.25$ | 0.64 | 0.56 | 0.46 | 0.38 | 0.30 | 0.23 | 0.15 | 0.10 | 0.00 | 0.00 |
| | $\sigma = 0.50$ | $\epsilon = 0.50$ | 0.63 | 0.55 | 0.47 | 0.39 | 0.30 | 0.24 | 0.19 | 0.14 | 0.00 | 0.00 |
| | $\sigma = 0.50$ | $\epsilon = 1.00$ | 0.57 | 0.52 | 0.46 | 0.41 | 0.33 | 0.28 | 0.23 | 0.18 | 0.00 | 0.00 |
| | $\sigma = 0.50$ | $\epsilon = 2.00$ | 0.47 | 0.45 | 0.41 | 0.39 | **0.35** | **0.31** | **0.26** | **0.23** | 0.00 | 0.00 |
| | $\sigma = 1.00$ | $\epsilon = 0.25$ | 0.48 | 0.42 | 0.36 | 0.30 | 0.25 | 0.21 | 0.17 | 0.14 | 0.12 | 0.09 |
| | $\sigma = 1.00$ | $\epsilon = 0.50$ | 0.47 | 0.41 | 0.37 | 0.31 | 0.27 | 0.22 | 0.20 | 0.17 | 0.14 | 0.12 |
| | $\sigma = 1.00$ | $\epsilon = 1.00$ | 0.46 | 0.41 | 0.37 | 0.33 | 0.28 | 0.24 | 0.22 | 0.18 | 0.16 | 0.14 |
| | $\sigma = 1.00$ | $\epsilon = 2.00$ | 0.39 | 0.37 | 0.34 | 0.30 | 0.27 | 0.25 | 0.22 | 0.20 | **0.18** | **0.15** |

Table 8: SMOOTHADV-ersarial training $T = 4$ steps, $m_{train} = 1$ sample.

| | $\ell_2$ RADIUS (CIFAR-10) | 0.0 | 0.25 | 0.5 | 0.75 | 1.0 | 1.25 | 1.5 | 1.75 | 2.0 | 2.25 |
|---|---|---|---|---|---|---|---|---|---|---|---|
| [6] | $\sigma = 0.12$ | 0.81 | 0.59 | 0.00 | 0.00 | 0.00 | 0.00 | 0.00 | 0.00 | 0.00 | 0.00 |
| | $\sigma = 0.25$ | 0.75 | 0.60 | 0.43 | 0.27 | 0.00 | 0.00 | 0.00 | 0.00 | 0.00 | 0.00 |
| | $\sigma = 0.50$ | 0.65 | 0.55 | 0.41 | 0.32 | 0.23 | 0.15 | 0.09 | 0.05 | 0.00 | 0.00 |
| | $\sigma = 1.00$ | 0.47 | 0.39 | 0.34 | 0.28 | 0.22 | 0.17 | 0.14 | 0.12 | 0.10 | 0.08 |
| SMOOTHADV$_{\text{PGD}}$ | $\sigma = 0.12$, $\epsilon = 0.25$ | 0.82 | 0.68 | 0.00 | 0.00 | 0.00 | 0.00 | 0.00 | 0.00 | 0.00 | 0.00 |
| | $\sigma = 0.12$, $\epsilon = 0.50$ | 0.80 | 0.67 | 0.00 | 0.00 | 0.00 | 0.00 | 0.00 | 0.00 | 0.00 | 0.00 |
| | $\sigma = 0.12$, $\epsilon = 1.00$ | 0.78 | **0.69** | 0.00 | 0.00 | 0.00 | 0.00 | 0.00 | 0.00 | 0.00 | 0.00 |
| | $\sigma = 0.12$, $\epsilon = 2.00$ | 0.78 | 0.67 | 0.00 | 0.00 | 0.00 | 0.00 | 0.00 | 0.00 | 0.00 | 0.00 |
| | $\sigma = 0.25$, $\epsilon = 0.25$ | 0.77 | 0.64 | 0.50 | 0.38 | 0.00 | 0.00 | 0.00 | 0.00 | 0.00 | 0.00 |
| | $\sigma = 0.25$, $\epsilon = 0.50$ | 0.70 | 0.61 | 0.50 | 0.40 | 0.00 | 0.00 | 0.00 | 0.00 | 0.00 | 0.00 |
| | $\sigma = 0.25$, $\epsilon = 1.00$ | 0.72 | 0.61 | 0.53 | 0.42 | 0.00 | 0.00 | 0.00 | 0.00 | 0.00 | 0.00 |
| | $\sigma = 0.25$, $\epsilon = 2.00$ | 0.72 | 0.63 | **0.54** | 0.40 | 0.00 | 0.00 | 0.00 | 0.00 | 0.00 | 0.00 |
| | $\sigma = 0.50$, $\epsilon = 0.25$ | 0.65 | 0.57 | 0.47 | 0.37 | 0.27 | 0.19 | 0.12 | 0.07 | 0.00 | 0.00 |
| | $\sigma = 0.50$, $\epsilon = 0.50$ | 0.64 | 0.54 | 0.45 | 0.35 | 0.28 | 0.20 | 0.15 | 0.10 | 0.00 | 0.00 |
| | $\sigma = 0.50$, $\epsilon = 1.00$ | 0.63 | 0.54 | 0.46 | 0.38 | 0.30 | 0.23 | 0.16 | 0.11 | 0.00 | 0.00 |
| | $\sigma = 0.50$, $\epsilon = 2.00$ | 0.63 | 0.53 | 0.44 | 0.36 | 0.29 | 0.22 | 0.17 | 0.10 | 0.00 | 0.00 |
| | $\sigma = 1.00$, $\epsilon = 0.25$ | 0.48 | 0.41 | 0.34 | 0.29 | 0.22 | 0.19 | 0.17 | 0.14 | 0.10 | 0.09 |
| | $\sigma = 1.00$, $\epsilon = 0.50$ | 0.47 | 0.40 | 0.34 | 0.28 | 0.23 | 0.20 | 0.17 | 0.14 | 0.11 | 0.09 |
| | $\sigma = 1.00$, $\epsilon = 1.00$ | 0.47 | 0.39 | 0.34 | 0.28 | 0.24 | 0.21 | 0.18 | 0.15 | 0.13 | 0.09 |
| | $\sigma = 1.00$, $\epsilon = 2.00$ | 0.48 | 0.40 | 0.35 | 0.30 | 0.25 | 0.21 | 0.17 | 0.14 | 0.12 | 0.09 |
| SMOOTHADV$_{\text{DDN}}$ | $\sigma = 0.12$, $\epsilon = 0.25$ | **0.83** | 0.69 | 0.00 | 0.00 | 0.00 | 0.00 | 0.00 | 0.00 | 0.00 | 0.00 |
| | $\sigma = 0.12$, $\epsilon = 0.50$ | 0.81 | 0.69 | 0.00 | 0.00 | 0.00 | 0.00 | 0.00 | 0.00 | 0.00 | 0.00 |
| | $\sigma = 0.12$, $\epsilon = 1.00$ | 0.72 | 0.63 | 0.00 | 0.00 | 0.00 | 0.00 | 0.00 | 0.00 | 0.00 | 0.00 |
| | $\sigma = 0.12$, $\epsilon = 2.00$ | 0.56 | 0.52 | 0.00 | 0.00 | 0.00 | 0.00 | 0.00 | 0.00 | 0.00 | 0.00 |
| | $\sigma = 0.25$, $\epsilon = 0.25$ | 0.76 | 0.66 | 0.51 | 0.39 | 0.00 | 0.00 | 0.00 | 0.00 | 0.00 | 0.00 |
| | $\sigma = 0.25$, $\epsilon = 0.50$ | 0.69 | 0.63 | 0.53 | 0.42 | 0.00 | 0.00 | 0.00 | 0.00 | 0.00 | 0.00 |
| | $\sigma = 0.25$, $\epsilon = 1.00$ | 0.66 | 0.59 | 0.53 | **0.46** | 0.00 | 0.00 | 0.00 | 0.00 | 0.00 | 0.00 |
| | $\sigma = 0.25$, $\epsilon = 2.00$ | 0.53 | 0.49 | 0.45 | 0.42 | 0.00 | 0.00 | 0.00 | 0.00 | 0.00 | 0.00 |
| | $\sigma = 0.50$, $\epsilon = 0.25$ | 0.65 | 0.57 | 0.47 | 0.37 | 0.29 | 0.23 | 0.16 | 0.09 | 0.00 | 0.00 |
| | $\sigma = 0.50$, $\epsilon = 0.50$ | 0.62 | 0.54 | 0.48 | 0.40 | 0.29 | 0.25 | 0.19 | 0.14 | 0.00 | 0.00 |
| | $\sigma = 0.50$, $\epsilon = 1.00$ | 0.56 | 0.50 | 0.44 | 0.39 | **0.34** | 0.30 | 0.23 | 0.18 | 0.00 | 0.00 |
| | $\sigma = 0.50$, $\epsilon = 2.00$ | 0.47 | 0.44 | 0.41 | 0.38 | 0.34 | **0.31** | **0.27** | **0.24** | 0.00 | 0.00 |
| | $\sigma = 1.00$, $\epsilon = 0.25$ | 0.49 | 0.42 | 0.36 | 0.30 | 0.25 | 0.21 | 0.18 | 0.14 | 0.12 | 0.10 |
| | $\sigma = 1.00$, $\epsilon = 0.50$ | 0.48 | 0.43 | 0.37 | 0.30 | 0.26 | 0.24 | 0.19 | 0.16 | 0.14 | 0.12 |
| | $\sigma = 1.00$, $\epsilon = 1.00$ | 0.45 | 0.40 | 0.37 | 0.34 | 0.30 | 0.25 | 0.21 | 0.19 | 0.17 | 0.15 |
| | $\sigma = 1.00$, $\epsilon = 2.00$ | 0.37 | 0.35 | 0.32 | 0.30 | 0.28 | 0.25 | 0.23 | 0.19 | **0.17** | **0.15** |

Table 9: SMOOTHADV-ersarial training $T = 6$ steps, $m_{train} = 1$ sample.

| ℓ₂ RADIUS (CIFAR-10) | | 0.0 | 0.25 | 0.5 | 0.75 | 1.0 | 1.25 | 1.5 | 1.75 | 2.0 | 2.25 |
|---|---|---|---|---|---|---|---|---|---|---|---|
| [6] $\sigma = 0.12$ | | **0.81** | 0.59 | 0.00 | 0.00 | 0.00 | 0.00 | 0.00 | 0.00 | 0.00 | 0.00 |
| $\sigma = 0.25$ | | 0.75 | 0.60 | 0.43 | 0.27 | 0.00 | 0.00 | 0.00 | 0.00 | 0.00 | 0.00 |
| $\sigma = 0.50$ | | 0.65 | 0.55 | 0.41 | 0.32 | 0.23 | 0.15 | 0.09 | 0.05 | 0.00 | 0.00 |
| $\sigma = 1.00$ | | 0.47 | 0.39 | 0.34 | 0.28 | 0.22 | 0.17 | 0.14 | 0.12 | 0.10 | 0.08 |
| SMOOTHADV_PGD $\sigma = 0.12$ | $\epsilon = 0.25$ | **0.81** | 0.66 | 0.00 | 0.00 | 0.00 | 0.00 | 0.00 | 0.00 | 0.00 | 0.00 |
| $\sigma = 0.12$ | $\epsilon = 0.50$ | **0.81** | 0.67 | 0.00 | 0.00 | 0.00 | 0.00 | 0.00 | 0.00 | 0.00 | 0.00 |
| $\sigma = 0.12$ | $\epsilon = 1.00$ | 0.76 | 0.66 | 0.00 | 0.00 | 0.00 | 0.00 | 0.00 | 0.00 | 0.00 | 0.00 |
| $\sigma = 0.12$ | $\epsilon = 2.00$ | 0.80 | 0.67 | 0.00 | 0.00 | 0.00 | 0.00 | 0.00 | 0.00 | 0.00 | 0.00 |
| $\sigma = 0.25$ | $\epsilon = 0.25$ | 0.77 | 0.65 | 0.49 | 0.36 | 0.00 | 0.00 | 0.00 | 0.00 | 0.00 | 0.00 |
| $\sigma = 0.25$ | $\epsilon = 0.50$ | 0.75 | 0.64 | 0.51 | 0.37 | 0.00 | 0.00 | 0.00 | 0.00 | 0.00 | 0.00 |
| $\sigma = 0.25$ | $\epsilon = 1.00$ | 0.72 | 0.63 | 0.53 | 0.41 | 0.00 | 0.00 | 0.00 | 0.00 | 0.00 | 0.00 |
| $\sigma = 0.25$ | $\epsilon = 2.00$ | 0.71 | 0.63 | 0.52 | 0.40 | 0.00 | 0.00 | 0.00 | 0.00 | 0.00 | 0.00 |
| $\sigma = 0.50$ | $\epsilon = 0.25$ | 0.68 | 0.56 | 0.47 | 0.36 | 0.25 | 0.19 | 0.12 | 0.08 | 0.00 | 0.00 |
| $\sigma = 0.50$ | $\epsilon = 0.50$ | 0.67 | 0.58 | 0.45 | 0.38 | 0.30 | 0.22 | 0.16 | 0.11 | 0.00 | 0.00 |
| $\sigma = 0.50$ | $\epsilon = 1.00$ | 0.62 | 0.52 | 0.43 | 0.35 | 0.29 | 0.25 | 0.18 | 0.12 | 0.00 | 0.00 |
| $\sigma = 0.50$ | $\epsilon = 2.00$ | 0.63 | 0.54 | 0.45 | 0.36 | 0.27 | 0.22 | 0.16 | 0.11 | 0.00 | 0.00 |
| $\sigma = 1.00$ | $\epsilon = 0.25$ | 0.48 | 0.41 | 0.35 | 0.30 | 0.23 | 0.19 | 0.15 | 0.12 | 0.10 | 0.08 |
| $\sigma = 1.00$ | $\epsilon = 0.50$ | 0.47 | 0.40 | 0.35 | 0.30 | 0.23 | 0.19 | 0.17 | 0.13 | 0.10 | 0.09 |
| $\sigma = 1.00$ | $\epsilon = 1.00$ | 0.47 | 0.40 | 0.35 | 0.30 | 0.24 | 0.21 | 0.17 | 0.15 | 0.13 | 0.09 |
| $\sigma = 1.00$ | $\epsilon = 2.00$ | 0.45 | 0.40 | 0.34 | 0.30 | 0.24 | 0.18 | 0.17 | 0.15 | 0.12 | 0.09 |
| SMOOTHADV_DDN $\sigma = 0.12$ | $\epsilon = 0.25$ | **0.81** | 0.65 | 0.00 | 0.00 | 0.00 | 0.00 | 0.00 | 0.00 | 0.00 | 0.00 |
| $\sigma = 0.12$ | $\epsilon = 0.50$ | 0.78 | **0.68** | 0.00 | 0.00 | 0.00 | 0.00 | 0.00 | 0.00 | 0.00 | 0.00 |
| $\sigma = 0.12$ | $\epsilon = 1.00$ | 0.70 | 0.62 | 0.00 | 0.00 | 0.00 | 0.00 | 0.00 | 0.00 | 0.00 | 0.00 |
| $\sigma = 0.12$ | $\epsilon = 2.00$ | 0.56 | 0.53 | 0.00 | 0.00 | 0.00 | 0.00 | 0.00 | 0.00 | 0.00 | 0.00 |
| $\sigma = 0.25$ | $\epsilon = 0.25$ | 0.76 | 0.63 | **0.54** | 0.40 | 0.00 | 0.00 | 0.00 | 0.00 | 0.00 | 0.00 |
| $\sigma = 0.25$ | $\epsilon = 0.50$ | 0.72 | 0.61 | 0.52 | 0.43 | 0.00 | 0.00 | 0.00 | 0.00 | 0.00 | 0.00 |
| $\sigma = 0.25$ | $\epsilon = 1.00$ | 0.61 | 0.57 | 0.52 | **0.45** | 0.00 | 0.00 | 0.00 | 0.00 | 0.00 | 0.00 |
| $\sigma = 0.25$ | $\epsilon = 2.00$ | 0.52 | 0.48 | 0.44 | 0.40 | 0.00 | 0.00 | 0.00 | 0.00 | 0.00 | 0.00 |
| $\sigma = 0.50$ | $\epsilon = 0.25$ | 0.65 | 0.56 | 0.45 | 0.36 | 0.29 | 0.22 | 0.14 | 0.09 | 0.00 | 0.00 |
| $\sigma = 0.50$ | $\epsilon = 0.50$ | 0.62 | 0.55 | 0.46 | 0.38 | 0.31 | 0.25 | 0.20 | 0.16 | 0.00 | 0.00 |
| $\sigma = 0.50$ | $\epsilon = 1.00$ | 0.55 | 0.51 | 0.45 | 0.40 | **0.35** | 0.29 | 0.24 | 0.19 | 0.00 | 0.00 |
| $\sigma = 0.50$ | $\epsilon = 2.00$ | 0.47 | 0.44 | 0.42 | 0.39 | 0.35 | **0.30** | **0.28** | **0.23** | 0.00 | 0.00 |
| $\sigma = 1.00$ | $\epsilon = 0.25$ | 0.47 | 0.41 | 0.36 | 0.30 | 0.25 | 0.21 | 0.18 | 0.14 | 0.12 | 0.10 |
| $\sigma = 1.00$ | $\epsilon = 0.50$ | 0.47 | 0.41 | 0.37 | 0.31 | 0.28 | 0.23 | 0.20 | 0.17 | 0.14 | 0.11 |
| $\sigma = 1.00$ | $\epsilon = 1.00$ | 0.45 | 0.40 | 0.36 | 0.32 | 0.29 | 0.26 | 0.22 | 0.19 | 0.15 | 0.14 |
| $\sigma = 1.00$ | $\epsilon = 2.00$ | 0.39 | 0.36 | 0.32 | 0.31 | 0.27 | 0.24 | 0.22 | 0.19 | **0.18** | **0.16** |

Table 10: SMOOTHADV-ersarial training $T = 8$ steps, $m_{train} = 1$ sample.

| $\ell_2$ RADIUS (CIFAR-10) | | 0.0 | 0.25 | 0.5 | 0.75 | 1.0 | 1.25 | 1.5 | 1.75 | 2.0 | 2.25 |
|---|---|---|---|---|---|---|---|---|---|---|---|
| [6] $\sigma = 0.12$ | | 0.81 | 0.59 | 0.00 | 0.00 | 0.00 | 0.00 | 0.00 | 0.00 | 0.00 | 0.00 |
| $\sigma = 0.25$ | | 0.75 | 0.60 | 0.43 | 0.27 | 0.00 | 0.00 | 0.00 | 0.00 | 0.00 | 0.00 |
| $\sigma = 0.50$ | | 0.65 | 0.55 | 0.41 | 0.32 | 0.23 | 0.15 | 0.09 | 0.05 | 0.00 | 0.00 |
| $\sigma = 1.00$ | | 0.47 | 0.39 | 0.34 | 0.28 | 0.22 | 0.17 | 0.14 | 0.12 | 0.10 | 0.08 |
| SMOOTHADV$_{\text{PGD}}$ $\sigma = 0.12$ | $\epsilon = 0.25$ | **0.83** | 0.68 | 0.00 | 0.00 | 0.00 | 0.00 | 0.00 | 0.00 | 0.00 | 0.00 |
| $\sigma = 0.12$ | $\epsilon = 0.50$ | 0.82 | **0.70** | 0.00 | 0.00 | 0.00 | 0.00 | 0.00 | 0.00 | 0.00 | 0.00 |
| $\sigma = 0.12$ | $\epsilon = 1.00$ | 0.77 | 0.66 | 0.00 | 0.00 | 0.00 | 0.00 | 0.00 | 0.00 | 0.00 | 0.00 |
| $\sigma = 0.12$ | $\epsilon = 2.00$ | 0.74 | 0.65 | 0.00 | 0.00 | 0.00 | 0.00 | 0.00 | 0.00 | 0.00 | 0.00 |
| $\sigma = 0.25$ | $\epsilon = 0.25$ | 0.76 | 0.64 | 0.50 | 0.37 | 0.00 | 0.00 | 0.00 | 0.00 | 0.00 | 0.00 |
| $\sigma = 0.25$ | $\epsilon = 0.50$ | 0.75 | 0.63 | 0.51 | 0.41 | 0.00 | 0.00 | 0.00 | 0.00 | 0.00 | 0.00 |
| $\sigma = 0.25$ | $\epsilon = 1.00$ | 0.72 | 0.64 | **0.56** | **0.44** | 0.00 | 0.00 | 0.00 | 0.00 | 0.00 | 0.00 |
| $\sigma = 0.25$ | $\epsilon = 2.00$ | 0.71 | 0.61 | 0.54 | 0.41 | 0.00 | 0.00 | 0.00 | 0.00 | 0.00 | 0.00 |
| $\sigma = 0.50$ | $\epsilon = 0.25$ | 0.67 | 0.56 | 0.45 | 0.34 | 0.26 | 0.18 | 0.12 | 0.07 | 0.00 | 0.00 |
| $\sigma = 0.50$ | $\epsilon = 0.50$ | 0.65 | 0.57 | 0.45 | 0.37 | 0.29 | 0.22 | 0.16 | 0.09 | 0.00 | 0.00 |
| $\sigma = 0.50$ | $\epsilon = 1.00$ | 0.60 | 0.53 | 0.43 | 0.35 | 0.30 | 0.24 | 0.18 | 0.13 | 0.00 | 0.00 |
| $\sigma = 0.50$ | $\epsilon = 2.00$ | 0.63 | 0.53 | 0.45 | 0.36 | 0.29 | 0.22 | 0.15 | 0.12 | 0.00 | 0.00 |
| $\sigma = 1.00$ | $\epsilon = 0.25$ | 0.47 | 0.41 | 0.35 | 0.28 | 0.22 | 0.19 | 0.17 | 0.12 | 0.10 | 0.08 |
| $\sigma = 1.00$ | $\epsilon = 0.50$ | 0.44 | 0.39 | 0.32 | 0.28 | 0.23 | 0.19 | 0.16 | 0.13 | 0.11 | 0.07 |
| $\sigma = 1.00$ | $\epsilon = 1.00$ | 0.47 | 0.39 | 0.34 | 0.30 | 0.25 | 0.20 | 0.18 | 0.14 | 0.11 | 0.09 |
| $\sigma = 1.00$ | $\epsilon = 2.00$ | 0.46 | 0.40 | 0.33 | 0.30 | 0.26 | 0.20 | 0.18 | 0.15 | 0.12 | 0.10 |
| SMOOTHADV$_{\text{DDN}}$ $\sigma = 0.12$ | $\epsilon = 0.25$ | 0.81 | 0.68 | 0.00 | 0.00 | 0.00 | 0.00 | 0.00 | 0.00 | 0.00 | 0.00 |
| $\sigma = 0.12$ | $\epsilon = 0.50$ | 0.77 | 0.66 | 0.00 | 0.00 | 0.00 | 0.00 | 0.00 | 0.00 | 0.00 | 0.00 |
| $\sigma = 0.12$ | $\epsilon = 1.00$ | 0.71 | 0.64 | 0.00 | 0.00 | 0.00 | 0.00 | 0.00 | 0.00 | 0.00 | 0.00 |
| $\sigma = 0.12$ | $\epsilon = 2.00$ | 0.56 | 0.52 | 0.00 | 0.00 | 0.00 | 0.00 | 0.00 | 0.00 | 0.00 | 0.00 |
| $\sigma = 0.25$ | $\epsilon = 0.25$ | 0.73 | 0.64 | 0.50 | 0.39 | 0.00 | 0.00 | 0.00 | 0.00 | 0.00 | 0.00 |
| $\sigma = 0.25$ | $\epsilon = 0.50$ | 0.74 | 0.63 | 0.53 | 0.41 | 0.00 | 0.00 | 0.00 | 0.00 | 0.00 | 0.00 |
| $\sigma = 0.25$ | $\epsilon = 1.00$ | 0.64 | 0.58 | 0.51 | **0.44** | 0.00 | 0.00 | 0.00 | 0.00 | 0.00 | 0.00 |
| $\sigma = 0.25$ | $\epsilon = 2.00$ | 0.52 | 0.48 | 0.44 | 0.40 | 0.00 | 0.00 | 0.00 | 0.00 | 0.00 | 0.00 |
| $\sigma = 0.50$ | $\epsilon = 0.25$ | 0.64 | 0.55 | 0.47 | 0.37 | 0.29 | 0.22 | 0.14 | 0.09 | 0.00 | 0.00 |
| $\sigma = 0.50$ | $\epsilon = 0.50$ | 0.63 | 0.57 | 0.48 | 0.39 | 0.32 | 0.26 | 0.19 | 0.13 | 0.00 | 0.00 |
| $\sigma = 0.50$ | $\epsilon = 1.00$ | 0.56 | 0.51 | 0.44 | 0.40 | **0.36** | **0.31** | 0.25 | 0.20 | 0.00 | 0.00 |
| $\sigma = 0.50$ | $\epsilon = 2.00$ | 0.46 | 0.43 | 0.40 | 0.38 | 0.35 | 0.31 | **0.26** | **0.22** | 0.00 | 0.00 |
| $\sigma = 1.00$ | $\epsilon = 0.25$ | 0.49 | 0.41 | 0.35 | 0.30 | 0.25 | 0.21 | 0.17 | 0.15 | 0.12 | 0.10 |
| $\sigma = 1.00$ | $\epsilon = 0.50$ | 0.48 | 0.42 | 0.36 | 0.31 | 0.26 | 0.22 | 0.19 | 0.16 | 0.14 | 0.12 |
| $\sigma = 1.00$ | $\epsilon = 1.00$ | 0.44 | 0.41 | 0.35 | 0.33 | 0.29 | 0.25 | 0.22 | 0.19 | 0.16 | 0.14 |
| $\sigma = 1.00$ | $\epsilon = 2.00$ | 0.39 | 0.36 | 0.34 | 0.30 | 0.28 | 0.25 | 0.23 | 0.21 | **0.18** | **0.16** |

Table 11: SMOOTHADV-ersarial training $T = 10$ steps, $m_{train} = 1$ sample.

| | $\ell_2$ RADIUS (CIFAR-10) | 0.0 | 0.25 | 0.5 | 0.75 | 1.0 | 1.25 | 1.5 | 1.75 | 2.0 | 2.25 |
|---|---|---|---|---|---|---|---|---|---|---|---|
| [6] | $\sigma = 0.12$ | 0.81 | 0.59 | 0.00 | 0.00 | 0.00 | 0.00 | 0.00 | 0.00 | 0.00 | 0.00 |
| | $\sigma = 0.25$ | 0.75 | 0.60 | 0.43 | 0.27 | 0.00 | 0.00 | 0.00 | 0.00 | 0.00 | 0.00 |
| | $\sigma = 0.50$ | 0.65 | 0.55 | 0.41 | 0.32 | 0.23 | 0.15 | 0.09 | 0.05 | 0.00 | 0.00 |
| | $\sigma = 1.00$ | 0.47 | 0.39 | 0.34 | 0.28 | 0.22 | 0.17 | 0.14 | 0.12 | 0.10 | 0.08 |
| SMOOTHADV$_{\text{PGD}}$ | $\sigma = 0.12$ $\epsilon = 0.25$ | **0.84** | 0.66 | 0.00 | 0.00 | 0.00 | 0.00 | 0.00 | 0.00 | 0.00 | 0.00 |
| | $\sigma = 0.12$ $\epsilon = 0.50$ | 0.81 | 0.66 | 0.00 | 0.00 | 0.00 | 0.00 | 0.00 | 0.00 | 0.00 | 0.00 |
| | $\sigma = 0.12$ $\epsilon = 1.00$ | 0.76 | 0.67 | 0.00 | 0.00 | 0.00 | 0.00 | 0.00 | 0.00 | 0.00 | 0.00 |
| | $\sigma = 0.12$ $\epsilon = 2.00$ | 0.75 | 0.64 | 0.00 | 0.00 | 0.00 | 0.00 | 0.00 | 0.00 | 0.00 | 0.00 |
| | $\sigma = 0.25$ $\epsilon = 0.25$ | 0.75 | 0.63 | 0.47 | 0.34 | 0.00 | 0.00 | 0.00 | 0.00 | 0.00 | 0.00 |
| | $\sigma = 0.25$ $\epsilon = 0.50$ | 0.73 | 0.64 | 0.52 | 0.41 | 0.00 | 0.00 | 0.00 | 0.00 | 0.00 | 0.00 |
| | $\sigma = 0.25$ $\epsilon = 1.00$ | 0.71 | 0.62 | **0.53** | 0.43 | 0.00 | 0.00 | 0.00 | 0.00 | 0.00 | 0.00 |
| | $\sigma = 0.25$ $\epsilon = 2.00$ | 0.71 | 0.62 | 0.51 | 0.41 | 0.00 | 0.00 | 0.00 | 0.00 | 0.00 | 0.00 |
| | $\sigma = 0.50$ $\epsilon = 0.25$ | 0.67 | 0.55 | 0.45 | 0.34 | 0.26 | 0.19 | 0.13 | 0.07 | 0.00 | 0.00 |
| | $\sigma = 0.50$ $\epsilon = 0.50$ | 0.65 | 0.57 | 0.45 | 0.38 | 0.29 | 0.23 | 0.17 | 0.10 | 0.00 | 0.00 |
| | $\sigma = 0.50$ $\epsilon = 1.00$ | 0.62 | 0.56 | 0.47 | 0.38 | 0.32 | 0.25 | 0.19 | 0.12 | 0.00 | 0.00 |
| | $\sigma = 0.50$ $\epsilon = 2.00$ | 0.64 | 0.53 | 0.45 | 0.37 | 0.29 | 0.22 | 0.16 | 0.11 | 0.00 | 0.00 |
| | $\sigma = 1.00$ $\epsilon = 0.25$ | 0.49 | 0.43 | 0.34 | 0.27 | 0.22 | 0.18 | 0.15 | 0.12 | 0.10 | 0.07 |
| | $\sigma = 1.00$ $\epsilon = 0.50$ | 0.48 | 0.42 | 0.36 | 0.29 | 0.24 | 0.19 | 0.16 | 0.14 | 0.11 | 0.09 |
| | $\sigma = 1.00$ $\epsilon = 1.00$ | 0.47 | 0.41 | 0.33 | 0.29 | 0.25 | 0.21 | 0.18 | 0.16 | 0.13 | 0.10 |
| | $\sigma = 1.00$ $\epsilon = 2.00$ | 0.48 | 0.40 | 0.36 | 0.30 | 0.24 | 0.21 | 0.17 | 0.13 | 0.12 | 0.10 |
| SMOOTHADV$_{\text{DDN}}$ | $\sigma = 0.12$ $\epsilon = 0.25$ | 0.82 | **0.69** | 0.00 | 0.00 | 0.00 | 0.00 | 0.00 | 0.00 | 0.00 | 0.00 |
| | $\sigma = 0.12$ $\epsilon = 0.50$ | 0.79 | 0.68 | 0.00 | 0.00 | 0.00 | 0.00 | 0.00 | 0.00 | 0.00 | 0.00 |
| | $\sigma = 0.12$ $\epsilon = 1.00$ | 0.72 | 0.62 | 0.00 | 0.00 | 0.00 | 0.00 | 0.00 | 0.00 | 0.00 | 0.00 |
| | $\sigma = 0.12$ $\epsilon = 2.00$ | 0.54 | 0.52 | 0.00 | 0.00 | 0.00 | 0.00 | 0.00 | 0.00 | 0.00 | 0.00 |
| | $\sigma = 0.25$ $\epsilon = 0.25$ | 0.74 | 0.63 | 0.50 | 0.39 | 0.00 | 0.00 | 0.00 | 0.00 | 0.00 | 0.00 |
| | $\sigma = 0.25$ $\epsilon = 0.50$ | 0.75 | 0.64 | **0.53** | 0.43 | 0.00 | 0.00 | 0.00 | 0.00 | 0.00 | 0.00 |
| | $\sigma = 0.25$ $\epsilon = 1.00$ | 0.63 | 0.56 | 0.51 | **0.46** | 0.00 | 0.00 | 0.00 | 0.00 | 0.00 | 0.00 |
| | $\sigma = 0.25$ $\epsilon = 2.00$ | 0.53 | 0.49 | 0.46 | 0.42 | 0.00 | 0.00 | 0.00 | 0.00 | 0.00 | 0.00 |
| | $\sigma = 0.50$ $\epsilon = 0.25$ | 0.62 | 0.54 | 0.44 | 0.36 | 0.27 | 0.22 | 0.14 | 0.10 | 0.00 | 0.00 |
| | $\sigma = 0.50$ $\epsilon = 0.50$ | 0.63 | 0.54 | 0.45 | 0.37 | 0.31 | 0.26 | 0.19 | 0.13 | 0.00 | 0.00 |
| | $\sigma = 0.50$ $\epsilon = 1.00$ | 0.55 | 0.51 | 0.45 | 0.38 | 0.33 | 0.28 | 0.24 | 0.18 | 0.00 | 0.00 |
| | $\sigma = 0.50$ $\epsilon = 2.00$ | 0.49 | 0.46 | 0.42 | 0.37 | **0.35** | **0.31** | **0.26** | **0.23** | 0.00 | 0.00 |
| | $\sigma = 1.00$ $\epsilon = 0.25$ | 0.49 | 0.41 | 0.36 | 0.31 | 0.25 | 0.22 | 0.18 | 0.15 | 0.12 | 0.10 |
| | $\sigma = 1.00$ $\epsilon = 0.50$ | 0.47 | 0.41 | 0.36 | 0.31 | 0.27 | 0.23 | 0.20 | 0.16 | 0.14 | 0.11 |
| | $\sigma = 1.00$ $\epsilon = 1.00$ | 0.46 | 0.43 | 0.38 | 0.33 | 0.29 | 0.26 | 0.22 | 0.18 | 0.15 | 0.14 |
| | $\sigma = 1.00$ $\epsilon = 2.00$ | 0.40 | 0.36 | 0.34 | 0.32 | 0.28 | 0.24 | 0.22 | 0.20 | **0.18** | **0.16** |

Table 12: SMOOTHADV$_{\mathrm{DDN}}$ training $T = 4$ steps, $m_{train} \in \{1, 2, 4, 8\}$ samples.

| | $\ell_2$ Radius (CIFAR-10) | | 0.0 | 0.25 | 0.5 | 0.75 | 1.0 | 1.25 | 1.5 | 1.75 | 2.0 | 2.25 |
|---|---|---|---|---|---|---|---|---|---|---|---|---|
| [6] | $\sigma = 0.12$ | | 0.81 | 0.59 | 0.00 | 0.00 | 0.00 | 0.00 | 0.00 | 0.00 | 0.00 | 0.00 |
| | $\sigma = 0.25$ | | 0.75 | 0.60 | 0.43 | 0.27 | 0.00 | 0.00 | 0.00 | 0.00 | 0.00 | 0.00 |
| | $\sigma = 0.50$ | | 0.65 | 0.55 | 0.41 | 0.32 | 0.23 | 0.15 | 0.09 | 0.05 | 0.00 | 0.00 |
| | $\sigma = 1.00$ | | 0.47 | 0.39 | 0.34 | 0.28 | 0.22 | 0.17 | 0.14 | 0.12 | 0.10 | 0.08 |
| $m_{train} = 1$ SAMPLE | $\sigma = 0.12$ | $\epsilon = 0.25$ | 0.82 | 0.68 | 0.00 | 0.00 | 0.00 | 0.00 | 0.00 | 0.00 | 0.00 | 0.00 |
| | $\sigma = 0.12$ | $\epsilon = 0.50$ | 0.80 | 0.67 | 0.00 | 0.00 | 0.00 | 0.00 | 0.00 | 0.00 | 0.00 | 0.00 |
| | $\sigma = 0.12$ | $\epsilon = 1.00$ | 0.78 | 0.69 | 0.00 | 0.00 | 0.00 | 0.00 | 0.00 | 0.00 | 0.00 | 0.00 |
| | $\sigma = 0.12$ | $\epsilon = 2.00$ | 0.78 | 0.67 | 0.00 | 0.00 | 0.00 | 0.00 | 0.00 | 0.00 | 0.00 | 0.00 |
| | $\sigma = 0.25$ | $\epsilon = 0.25$ | 0.77 | 0.64 | 0.50 | 0.38 | 0.00 | 0.00 | 0.00 | 0.00 | 0.00 | 0.00 |
| | $\sigma = 0.25$ | $\epsilon = 0.50$ | 0.70 | 0.61 | 0.50 | 0.40 | 0.00 | 0.00 | 0.00 | 0.00 | 0.00 | 0.00 |
| | $\sigma = 0.25$ | $\epsilon = 1.00$ | 0.72 | 0.61 | 0.53 | 0.42 | 0.00 | 0.00 | 0.00 | 0.00 | 0.00 | 0.00 |
| | $\sigma = 0.25$ | $\epsilon = 2.00$ | 0.72 | 0.63 | 0.54 | 0.40 | 0.00 | 0.00 | 0.00 | 0.00 | 0.00 | 0.00 |
| | $\sigma = 0.50$ | $\epsilon = 0.25$ | 0.65 | 0.57 | 0.47 | 0.37 | 0.27 | 0.19 | 0.12 | 0.07 | 0.00 | 0.00 |
| | $\sigma = 0.50$ | $\epsilon = 0.50$ | 0.64 | 0.54 | 0.45 | 0.35 | 0.28 | 0.20 | 0.15 | 0.10 | 0.00 | 0.00 |
| | $\sigma = 0.50$ | $\epsilon = 1.00$ | 0.63 | 0.54 | 0.46 | 0.38 | 0.30 | 0.23 | 0.16 | 0.11 | 0.00 | 0.00 |
| | $\sigma = 0.50$ | $\epsilon = 2.00$ | 0.63 | 0.53 | 0.44 | 0.36 | 0.29 | 0.22 | 0.17 | 0.10 | 0.00 | 0.00 |
| | $\sigma = 1.00$ | $\epsilon = 0.25$ | 0.48 | 0.41 | 0.34 | 0.29 | 0.22 | 0.19 | 0.17 | 0.14 | 0.10 | 0.09 |
| | $\sigma = 1.00$ | $\epsilon = 0.50$ | 0.47 | 0.40 | 0.34 | 0.28 | 0.23 | 0.20 | 0.17 | 0.14 | 0.11 | 0.09 |
| | $\sigma = 1.00$ | $\epsilon = 1.00$ | 0.47 | 0.39 | 0.34 | 0.28 | 0.24 | 0.21 | 0.18 | **0.15** | **0.13** | 0.09 |
| | $\sigma = 1.00$ | $\epsilon = 2.00$ | 0.48 | 0.40 | 0.35 | 0.30 | 0.25 | 0.21 | 0.17 | 0.14 | 0.12 | 0.09 |
| $m_{train} = 2$ SAMPLES | $\sigma = 0.12$ | $\epsilon = 0.25$ | 0.84 | 0.69 | 0.00 | 0.00 | 0.00 | 0.00 | 0.00 | 0.00 | 0.00 | 0.00 |
| | $\sigma = 0.12$ | $\epsilon = 0.50$ | 0.83 | 0.70 | 0.00 | 0.00 | 0.00 | 0.00 | 0.00 | 0.00 | 0.00 | 0.00 |
| | $\sigma = 0.12$ | $\epsilon = 1.00$ | 0.78 | 0.67 | 0.00 | 0.00 | 0.00 | 0.00 | 0.00 | 0.00 | 0.00 | 0.00 |
| | $\sigma = 0.12$ | $\epsilon = 2.00$ | 0.79 | 0.68 | 0.00 | 0.00 | 0.00 | 0.00 | 0.00 | 0.00 | 0.00 | 0.00 |
| | $\sigma = 0.25$ | $\epsilon = 0.25$ | 0.77 | 0.65 | 0.51 | 0.35 | 0.00 | 0.00 | 0.00 | 0.00 | 0.00 | 0.00 |
| | $\sigma = 0.25$ | $\epsilon = 0.50$ | 0.77 | 0.64 | 0.53 | 0.43 | 0.00 | 0.00 | 0.00 | 0.00 | 0.00 | 0.00 |
| | $\sigma = 0.25$ | $\epsilon = 1.00$ | 0.76 | 0.66 | 0.54 | 0.42 | 0.00 | 0.00 | 0.00 | 0.00 | 0.00 | 0.00 |
| | $\sigma = 0.25$ | $\epsilon = 2.00$ | 0.73 | 0.64 | 0.54 | 0.44 | 0.00 | 0.00 | 0.00 | 0.00 | 0.00 | 0.00 |
| | $\sigma = 0.50$ | $\epsilon = 0.25$ | 0.67 | 0.56 | 0.47 | 0.37 | 0.26 | 0.18 | 0.10 | 0.07 | 0.00 | 0.00 |
| | $\sigma = 0.50$ | $\epsilon = 0.50$ | 0.65 | 0.58 | 0.47 | 0.36 | 0.29 | 0.22 | 0.13 | 0.09 | 0.00 | 0.00 |
| | $\sigma = 0.50$ | $\epsilon = 1.00$ | 0.64 | 0.58 | 0.48 | 0.39 | 0.30 | 0.24 | 0.16 | 0.11 | 0.00 | 0.00 |
| | $\sigma = 0.50$ | $\epsilon = 2.00$ | 0.66 | 0.57 | 0.49 | 0.39 | **0.31** | **0.25** | 0.18 | 0.13 | 0.00 | 0.00 |
| | $\sigma = 1.00$ | $\epsilon = 0.25$ | 0.48 | 0.43 | 0.36 | 0.29 | 0.22 | 0.18 | 0.15 | 0.12 | 0.09 | 0.07 |
| | $\sigma = 1.00$ | $\epsilon = 0.50$ | 0.49 | 0.43 | 0.35 | 0.29 | 0.24 | 0.19 | 0.16 | 0.13 | 0.10 | 0.07 |
| | $\sigma = 1.00$ | $\epsilon = 1.00$ | 0.49 | 0.44 | 0.36 | 0.30 | 0.25 | 0.20 | 0.18 | **0.15** | 0.11 | 0.09 |
| | $\sigma = 1.00$ | $\epsilon = 2.00$ | 0.49 | 0.43 | 0.35 | 0.30 | 0.26 | 0.21 | **0.19** | 0.14 | 0.12 | 0.09 |
| $m_{train} = 4$ SAMPLES | $\sigma = 0.12$ | $\epsilon = 0.25$ | 0.85 | 0.71 | 0.00 | 0.00 | 0.00 | 0.00 | 0.00 | 0.00 | 0.00 | 0.00 |
| | $\sigma = 0.12$ | $\epsilon = 0.50$ | **0.85** | 0.72 | 0.00 | 0.00 | 0.00 | 0.00 | 0.00 | 0.00 | 0.00 | 0.00 |
| | $\sigma = 0.12$ | $\epsilon = 1.00$ | 0.81 | 0.72 | 0.00 | 0.00 | 0.00 | 0.00 | 0.00 | 0.00 | 0.00 | 0.00 |
| | $\sigma = 0.12$ | $\epsilon = 2.00$ | 0.82 | **0.73** | 0.00 | 0.00 | 0.00 | 0.00 | 0.00 | 0.00 | 0.00 | 0.00 |
| | $\sigma = 0.25$ | $\epsilon = 0.25$ | 0.79 | 0.66 | 0.51 | 0.35 | 0.00 | 0.00 | 0.00 | 0.00 | 0.00 | 0.00 |
| | $\sigma = 0.25$ | $\epsilon = 0.50$ | 0.79 | 0.63 | 0.51 | 0.39 | 0.00 | 0.00 | 0.00 | 0.00 | 0.00 | 0.00 |
| | $\sigma = 0.25$ | $\epsilon = 1.00$ | 0.77 | 0.68 | **0.56** | 0.42 | 0.00 | 0.00 | 0.00 | 0.00 | 0.00 | 0.00 |
| | $\sigma = 0.25$ | $\epsilon = 2.00$ | 0.76 | 0.68 | 0.56 | **0.45** | 0.00 | 0.00 | 0.00 | 0.00 | 0.00 | 0.00 |
| | $\sigma = 0.50$ | $\epsilon = 0.25$ | 0.68 | 0.57 | 0.46 | 0.37 | 0.27 | 0.17 | 0.11 | 0.08 | 0.00 | 0.00 |
| | $\sigma = 0.50$ | $\epsilon = 0.50$ | 0.68 | 0.59 | 0.48 | 0.37 | 0.28 | 0.21 | 0.14 | 0.08 | 0.00 | 0.00 |
| | $\sigma = 0.50$ | $\epsilon = 1.00$ | 0.66 | 0.58 | 0.47 | 0.38 | 0.31 | 0.24 | 0.17 | 0.11 | 0.00 | 0.00 |
| | $\sigma = 0.50$ | $\epsilon = 2.00$ | 0.65 | 0.58 | 0.49 | 0.40 | 0.29 | 0.23 | 0.17 | 0.10 | 0.00 | 0.00 |
| | $\sigma = 1.00$ | $\epsilon = 0.25$ | 0.49 | 0.41 | 0.35 | 0.29 | 0.22 | 0.19 | 0.15 | 0.12 | 0.09 | 0.07 |
| | $\sigma = 1.00$ | $\epsilon = 0.50$ | 0.50 | 0.44 | 0.36 | 0.29 | 0.24 | 0.19 | 0.16 | 0.12 | 0.09 | 0.06 |
| | $\sigma = 1.00$ | $\epsilon = 1.00$ | 0.48 | 0.42 | 0.35 | 0.30 | 0.24 | 0.19 | 0.16 | 0.13 | 0.10 | 0.08 |
| | $\sigma = 1.00$ | $\epsilon = 2.00$ | 0.39 | 0.34 | 0.28 | 0.23 | 0.18 | 0.15 | 0.13 | 0.09 | 0.08 | 0.06 |
| $m_{train} = 8$ SAMPLES | $\sigma = 0.12$ | $\epsilon = 0.50$ | 0.82 | 0.70 | 0.00 | 0.00 | 0.00 | 0.00 | 0.00 | 0.00 | 0.00 | 0.00 |
| | $\sigma = 0.12$ | $\epsilon = 1.00$ | 0.80 | 0.69 | 0.00 | 0.00 | 0.00 | 0.00 | 0.00 | 0.00 | 0.00 | 0.00 |
| | $\sigma = 0.12$ | $\epsilon = 2.00$ | 0.77 | 0.70 | 0.00 | 0.00 | 0.00 | 0.00 | 0.00 | 0.00 | 0.00 | 0.00 |
| | $\sigma = 0.25$ | $\epsilon = 0.25$ | 0.79 | 0.65 | 0.51 | 0.37 | 0.00 | 0.00 | 0.00 | 0.00 | 0.00 | 0.00 |
| | $\sigma = 0.25$ | $\epsilon = 0.50$ | 0.78 | 0.66 | 0.51 | 0.42 | 0.00 | 0.00 | 0.00 | 0.00 | 0.00 | 0.00 |
| | $\sigma = 0.25$ | $\epsilon = 1.00$ | 0.74 | 0.63 | 0.54 | 0.42 | 0.00 | 0.00 | 0.00 | 0.00 | 0.00 | 0.00 |
| | $\sigma = 0.25$ | $\epsilon = 2.00$ | 0.76 | 0.67 | 0.54 | 0.41 | 0.00 | 0.00 | 0.00 | 0.00 | 0.00 | 0.00 |
| | $\sigma = 0.50$ | $\epsilon = 0.25$ | 0.67 | 0.57 | 0.46 | 0.38 | 0.27 | 0.20 | 0.13 | 0.07 | 0.00 | 0.00 |
| | $\sigma = 0.50$ | $\epsilon = 0.50$ | 0.65 | 0.57 | 0.47 | 0.39 | 0.28 | 0.20 | 0.14 | 0.08 | 0.00 | 0.00 |
| | $\sigma = 0.50$ | $\epsilon = 1.00$ | 0.62 | 0.55 | 0.46 | 0.37 | 0.30 | 0.24 | 0.17 | 0.10 | 0.00 | 0.00 |
| | $\sigma = 0.50$ | $\epsilon = 2.00$ | 0.66 | 0.55 | 0.46 | 0.38 | 0.30 | 0.23 | 0.17 | 0.11 | 0.00 | 0.00 |
| | $\sigma = 1.00$ | $\epsilon = 0.25$ | 0.49 | 0.42 | 0.35 | 0.29 | 0.23 | 0.18 | 0.14 | 0.11 | 0.09 | 0.07 |
| | $\sigma = 1.00$ | $\epsilon = 0.50$ | 0.48 | 0.43 | 0.36 | 0.30 | 0.24 | 0.18 | 0.14 | 0.12 | 0.09 | 0.07 |
| | $\sigma = 1.00$ | $\epsilon = 1.00$ | 0.48 | 0.42 | 0.34 | 0.28 | 0.24 | 0.20 | 0.17 | 0.14 | 0.12 | 0.08 |
| | $\sigma = 1.00$ | $\epsilon = 2.00$ | 0.45 | 0.39 | 0.32 | 0.28 | 0.24 | 0.21 | 0.17 | 0.13 | 0.10 | 0.07 |

Table 13: SMOOTHADV$_{\text{DDN}}$ training $T = 10$ steps, $m_{train} \in \{1, 2, 4, 8\}$ samples.

| | $\ell_2$ RADIUS (CIFAR-10) | 0.0 | 0.25 | 0.5 | 0.75 | 1.0 | 1.25 | 1.5 | 1.75 | 2.0 | 2.25 |
|---|---|---|---|---|---|---|---|---|---|---|---|
| [6] | $\sigma = 0.12$ | 0.81 | 0.59 | 0.00 | 0.00 | 0.00 | 0.00 | 0.00 | 0.00 | 0.00 | 0.00 |
| | $\sigma = 0.25$ | 0.75 | 0.60 | 0.43 | 0.27 | 0.00 | 0.00 | 0.00 | 0.00 | 0.00 | 0.00 |
| | $\sigma = 0.50$ | 0.65 | 0.55 | 0.41 | 0.32 | 0.23 | 0.15 | 0.09 | 0.05 | 0.00 | 0.00 |
| | $\sigma = 1.00$ | 0.47 | 0.39 | 0.34 | 0.28 | 0.22 | 0.17 | 0.14 | 0.12 | 0.10 | 0.08 |
| $m_{train} = 1$ SAMPLE | $\sigma = 0.12$ $\epsilon = 0.25$ | 0.84 | 0.66 | 0.00 | 0.00 | 0.00 | 0.00 | 0.00 | 0.00 | 0.00 | 0.00 |
| | $\sigma = 0.12$ $\epsilon = 0.50$ | 0.81 | 0.66 | 0.00 | 0.00 | 0.00 | 0.00 | 0.00 | 0.00 | 0.00 | 0.00 |
| | $\sigma = 0.12$ $\epsilon = 1.00$ | 0.76 | 0.67 | 0.00 | 0.00 | 0.00 | 0.00 | 0.00 | 0.00 | 0.00 | 0.00 |
| | $\sigma = 0.12$ $\epsilon = 2.00$ | 0.75 | 0.64 | 0.00 | 0.00 | 0.00 | 0.00 | 0.00 | 0.00 | 0.00 | 0.00 |
| | $\sigma = 0.25$ $\epsilon = 0.25$ | 0.75 | 0.63 | 0.47 | 0.34 | 0.00 | 0.00 | 0.00 | 0.00 | 0.00 | 0.00 |
| | $\sigma = 0.25$ $\epsilon = 0.50$ | 0.73 | 0.64 | 0.52 | 0.41 | 0.00 | 0.00 | 0.00 | 0.00 | 0.00 | 0.00 |
| | $\sigma = 0.25$ $\epsilon = 1.00$ | 0.71 | 0.62 | 0.53 | 0.43 | 0.00 | 0.00 | 0.00 | 0.00 | 0.00 | 0.00 |
| | $\sigma = 0.25$ $\epsilon = 2.00$ | 0.71 | 0.62 | 0.51 | 0.41 | 0.00 | 0.00 | 0.00 | 0.00 | 0.00 | 0.00 |
| | $\sigma = 0.50$ $\epsilon = 0.25$ | 0.67 | 0.55 | 0.45 | 0.34 | 0.26 | 0.19 | 0.13 | 0.07 | 0.00 | 0.00 |
| | $\sigma = 0.50$ $\epsilon = 0.50$ | 0.65 | 0.57 | 0.45 | 0.38 | 0.29 | 0.23 | 0.17 | 0.10 | 0.00 | 0.00 |
| | $\sigma = 0.50$ $\epsilon = 1.00$ | 0.62 | 0.56 | 0.47 | 0.38 | 0.32 | 0.25 | **0.19** | 0.12 | 0.00 | 0.00 |
| | $\sigma = 0.50$ $\epsilon = 2.00$ | 0.64 | 0.53 | 0.45 | 0.37 | 0.29 | 0.22 | 0.16 | 0.11 | 0.00 | 0.00 |
| | $\sigma = 1.00$ $\epsilon = 0.25$ | 0.49 | 0.43 | 0.34 | 0.27 | 0.22 | 0.18 | 0.15 | 0.12 | 0.10 | 0.07 |
| | $\sigma = 1.00$ $\epsilon = 0.50$ | 0.48 | 0.42 | 0.36 | 0.29 | 0.24 | 0.19 | 0.16 | 0.14 | 0.11 | 0.09 |
| | $\sigma = 1.00$ $\epsilon = 1.00$ | 0.47 | 0.41 | 0.33 | 0.29 | 0.25 | 0.21 | 0.18 | **0.16** | **0.13** | **0.10** |
| | $\sigma = 1.00$ $\epsilon = 2.00$ | 0.48 | 0.40 | 0.36 | 0.30 | 0.24 | 0.21 | 0.17 | 0.13 | 0.12 | 0.10 |
| $m_{train} = 2$ SAMPLES | $\sigma = 0.12$ $\epsilon = 0.25$ | 0.86 | 0.71 | 0.00 | 0.00 | 0.00 | 0.00 | 0.00 | 0.00 | 0.00 | 0.00 |
| | $\sigma = 0.12$ $\epsilon = 0.50$ | 0.81 | 0.69 | 0.00 | 0.00 | 0.00 | 0.00 | 0.00 | 0.00 | 0.00 | 0.00 |
| | $\sigma = 0.12$ $\epsilon = 1.00$ | 0.78 | 0.67 | 0.00 | 0.00 | 0.00 | 0.00 | 0.00 | 0.00 | 0.00 | 0.00 |
| | $\sigma = 0.12$ $\epsilon = 2.00$ | 0.79 | 0.71 | 0.00 | 0.00 | 0.00 | 0.00 | 0.00 | 0.00 | 0.00 | 0.00 |
| | $\sigma = 0.25$ $\epsilon = 0.25$ | 0.77 | 0.63 | 0.49 | 0.34 | 0.00 | 0.00 | 0.00 | 0.00 | 0.00 | 0.00 |
| | $\sigma = 0.25$ $\epsilon = 1.00$ | 0.72 | 0.63 | 0.55 | 0.44 | 0.00 | 0.00 | 0.00 | 0.00 | 0.00 | 0.00 |
| | $\sigma = 0.25$ $\epsilon = 2.00$ | 0.74 | 0.65 | 0.55 | 0.44 | 0.00 | 0.00 | 0.00 | 0.00 | 0.00 | 0.00 |
| | $\sigma = 0.50$ $\epsilon = 0.25$ | 0.65 | 0.53 | 0.43 | 0.34 | 0.24 | 0.16 | 0.10 | 0.06 | 0.00 | 0.00 |
| | $\sigma = 0.50$ $\epsilon = 0.50$ | 0.65 | 0.58 | 0.47 | 0.37 | 0.29 | 0.20 | 0.13 | 0.08 | 0.00 | 0.00 |
| | $\sigma = 0.50$ $\epsilon = 1.00$ | 0.64 | 0.56 | 0.46 | 0.38 | 0.30 | 0.23 | 0.18 | 0.12 | 0.00 | 0.00 |
| | $\sigma = 0.50$ $\epsilon = 2.00$ | 0.63 | 0.55 | 0.49 | 0.41 | 0.33 | 0.24 | **0.19** | 0.12 | 0.00 | 0.00 |
| | $\sigma = 1.00$ $\epsilon = 0.50$ | 0.49 | 0.42 | 0.36 | 0.28 | 0.25 | 0.21 | 0.17 | 0.13 | 0.12 | 0.08 |
| | $\sigma = 1.00$ $\epsilon = 2.00$ | 0.49 | 0.43 | 0.38 | 0.30 | 0.26 | 0.22 | **0.19** | 0.15 | **0.13** | 0.09 |
| $m_{train} = 4$ SAMPLES | $\sigma = 0.12$ $\epsilon = 0.25$ | 0.86 | 0.72 | 0.00 | 0.00 | 0.00 | 0.00 | 0.00 | 0.00 | 0.00 | 0.00 |
| | $\sigma = 0.12$ $\epsilon = 0.50$ | 0.86 | **0.73** | 0.00 | 0.00 | 0.00 | 0.00 | 0.00 | 0.00 | 0.00 | 0.00 |
| | $\sigma = 0.12$ $\epsilon = 1.00$ | 0.81 | 0.71 | 0.00 | 0.00 | 0.00 | 0.00 | 0.00 | 0.00 | 0.00 | 0.00 |
| | $\sigma = 0.12$ $\epsilon = 2.00$ | 0.82 | 0.71 | 0.00 | 0.00 | 0.00 | 0.00 | 0.00 | 0.00 | 0.00 | 0.00 |
| | $\sigma = 0.25$ $\epsilon = 0.25$ | 0.79 | 0.65 | 0.48 | 0.34 | 0.00 | 0.00 | 0.00 | 0.00 | 0.00 | 0.00 |
| | $\sigma = 0.25$ $\epsilon = 0.50$ | 0.79 | 0.67 | 0.53 | 0.41 | 0.00 | 0.00 | 0.00 | 0.00 | 0.00 | 0.00 |
| | $\sigma = 0.25$ $\epsilon = 1.00$ | 0.77 | 0.66 | **0.57** | **0.45** | 0.00 | 0.00 | 0.00 | 0.00 | 0.00 | 0.00 |
| | $\sigma = 0.25$ $\epsilon = 2.00$ | 0.75 | 0.65 | 0.55 | 0.45 | 0.00 | 0.00 | 0.00 | 0.00 | 0.00 | 0.00 |
| | $\sigma = 0.50$ $\epsilon = 0.25$ | 0.67 | 0.59 | 0.47 | 0.36 | 0.27 | 0.19 | 0.14 | 0.09 | 0.00 | 0.00 |
| | $\sigma = 0.50$ $\epsilon = 0.50$ | 0.67 | 0.59 | 0.47 | 0.39 | 0.29 | 0.23 | 0.15 | 0.09 | 0.00 | 0.00 |
| | $\sigma = 0.50$ $\epsilon = 1.00$ | 0.66 | 0.58 | 0.50 | 0.42 | 0.33 | 0.25 | 0.17 | 0.10 | 0.00 | 0.00 |
| | $\sigma = 0.50$ $\epsilon = 2.00$ | 0.65 | 0.57 | 0.49 | 0.39 | **0.34** | **0.26** | 0.18 | 0.12 | 0.00 | 0.00 |
| | $\sigma = 1.00$ $\epsilon = 0.25$ | 0.48 | 0.41 | 0.35 | 0.28 | 0.23 | 0.19 | 0.16 | 0.13 | 0.09 | 0.07 |
| | $\sigma = 1.00$ $\epsilon = 0.50$ | 0.48 | 0.42 | 0.37 | 0.29 | 0.24 | 0.19 | 0.15 | 0.12 | 0.08 | 0.06 |
| | $\sigma = 1.00$ $\epsilon = 1.00$ | 0.49 | 0.43 | 0.36 | 0.30 | 0.25 | 0.21 | 0.16 | 0.13 | 0.12 | 0.08 |
| | $\sigma = 1.00$ $\epsilon = 2.00$ | 0.49 | 0.42 | 0.36 | 0.30 | 0.26 | 0.21 | 0.17 | 0.14 | 0.12 | 0.09 |
| $m_{train} = 8$ SAMPLES | $\sigma = 0.12$ $\epsilon = 0.25$ | **0.87** | 0.70 | 0.00 | 0.00 | 0.00 | 0.00 | 0.00 | 0.00 | 0.00 | 0.00 |
| | $\sigma = 0.12$ $\epsilon = 0.50$ | 0.86 | 0.71 | 0.00 | 0.00 | 0.00 | 0.00 | 0.00 | 0.00 | 0.00 | 0.00 |
| | $\sigma = 0.12$ $\epsilon = 1.00$ | 0.82 | 0.71 | 0.00 | 0.00 | 0.00 | 0.00 | 0.00 | 0.00 | 0.00 | 0.00 |
| | $\sigma = 0.12$ $\epsilon = 2.00$ | 0.81 | 0.70 | 0.00 | 0.00 | 0.00 | 0.00 | 0.00 | 0.00 | 0.00 | 0.00 |
| | $\sigma = 0.25$ $\epsilon = 0.25$ | 0.64 | 0.52 | 0.38 | 0.23 | 0.00 | 0.00 | 0.00 | 0.00 | 0.00 | 0.00 |
| | $\sigma = 0.25$ $\epsilon = 0.50$ | 0.80 | 0.69 | 0.53 | 0.37 | 0.00 | 0.00 | 0.00 | 0.00 | 0.00 | 0.00 |
| | $\sigma = 0.25$ $\epsilon = 1.00$ | 0.62 | 0.52 | 0.40 | 0.32 | 0.00 | 0.00 | 0.00 | 0.00 | 0.00 | 0.00 |
| | $\sigma = 0.25$ $\epsilon = 2.00$ | 0.71 | 0.62 | 0.52 | 0.43 | 0.00 | 0.00 | 0.00 | 0.00 | 0.00 | 0.00 |
| | $\sigma = 0.50$ $\epsilon = 0.25$ | 0.69 | 0.59 | 0.45 | 0.32 | 0.24 | 0.19 | 0.12 | 0.06 | 0.00 | 0.00 |
| | $\sigma = 0.50$ $\epsilon = 0.50$ | 0.65 | 0.55 | 0.47 | 0.39 | 0.31 | 0.23 | 0.16 | 0.10 | 0.00 | 0.00 |
| | $\sigma = 0.50$ $\epsilon = 1.00$ | 0.65 | 0.58 | 0.47 | 0.37 | 0.29 | 0.22 | 0.15 | 0.09 | 0.00 | 0.00 |
| | $\sigma = 0.50$ $\epsilon = 2.00$ | 0.65 | 0.56 | 0.46 | 0.36 | 0.29 | 0.22 | 0.16 | 0.10 | 0.00 | 0.00 |
| | $\sigma = 1.00$ $\epsilon = 0.25$ | 0.47 | 0.43 | 0.36 | 0.30 | 0.26 | 0.22 | 0.17 | 0.13 | 0.10 | 0.08 |
| | $\sigma = 1.00$ $\epsilon = 0.50$ | 0.48 | 0.42 | 0.35 | 0.28 | 0.22 | 0.18 | 0.16 | 0.13 | 0.10 | 0.07 |
| | $\sigma = 1.00$ $\epsilon = 1.00$ | 0.46 | 0.40 | 0.34 | 0.29 | 0.23 | 0.18 | 0.15 | 0.13 | 0.11 | 0.09 |
| | $\sigma = 1.00$ $\epsilon = 2.00$ | 0.46 | 0.40 | 0.34 | 0.29 | 0.24 | 0.20 | 0.17 | 0.15 | 0.11 | 0.09 |

Table 14: SMOOTHADV$_{\text{PGD}}$ training $T = 2$ steps, $m_{train} \in \{1, 2, 4, 8\}$ samples.

| | $\ell_2$ RADIUS (CIFAR-10) | | 0.0 | 0.25 | 0.5 | 0.75 | 1.0 | 1.25 | 1.5 | 1.75 | 2.0 | 2.25 |
|---|---|---|---|---|---|---|---|---|---|---|---|---|
| [6] | $\sigma = 0.12$ | | 0.81 | 0.59 | 0.00 | 0.00 | 0.00 | 0.00 | 0.00 | 0.00 | 0.00 | 0.00 |
| | $\sigma = 0.25$ | | 0.75 | 0.60 | 0.43 | 0.27 | 0.00 | 0.00 | 0.00 | 0.00 | 0.00 | 0.00 |
| | $\sigma = 0.50$ | | 0.65 | 0.55 | 0.41 | 0.32 | 0.23 | 0.15 | 0.09 | 0.05 | 0.00 | 0.00 |
| | $\sigma = 1.00$ | | 0.47 | 0.39 | 0.34 | 0.28 | 0.22 | 0.17 | 0.14 | 0.12 | 0.10 | 0.08 |
| $m_{train} = 1$ SAMPLE | $\sigma = 0.12$ | $\epsilon = 0.25$ | 0.82 | 0.69 | 0.00 | 0.00 | 0.00 | 0.00 | 0.00 | 0.00 | 0.00 | 0.00 |
| | $\sigma = 0.12$ | $\epsilon = 0.50$ | 0.79 | 0.67 | 0.00 | 0.00 | 0.00 | 0.00 | 0.00 | 0.00 | 0.00 | 0.00 |
| | $\sigma = 0.12$ | $\epsilon = 1.00$ | 0.71 | 0.64 | 0.00 | 0.00 | 0.00 | 0.00 | 0.00 | 0.00 | 0.00 | 0.00 |
| | $\sigma = 0.12$ | $\epsilon = 2.00$ | 0.54 | 0.49 | 0.00 | 0.00 | 0.00 | 0.00 | 0.00 | 0.00 | 0.00 | 0.00 |
| | $\sigma = 0.25$ | $\epsilon = 0.25$ | 0.77 | 0.65 | 0.51 | 0.39 | 0.00 | 0.00 | 0.00 | 0.00 | 0.00 | 0.00 |
| | $\sigma = 0.25$ | $\epsilon = 0.50$ | 0.74 | 0.64 | 0.53 | 0.41 | 0.00 | 0.00 | 0.00 | 0.00 | 0.00 | 0.00 |
| | $\sigma = 0.25$ | $\epsilon = 1.00$ | 0.64 | 0.59 | 0.53 | 0.45 | 0.00 | 0.00 | 0.00 | 0.00 | 0.00 | 0.00 |
| | $\sigma = 0.25$ | $\epsilon = 2.00$ | 0.53 | 0.49 | 0.46 | 0.42 | 0.00 | 0.00 | 0.00 | 0.00 | 0.00 | 0.00 |
| | $\sigma = 0.50$ | $\epsilon = 0.25$ | 0.64 | 0.56 | 0.46 | 0.38 | 0.30 | 0.23 | 0.15 | 0.10 | 0.00 | 0.00 |
| | $\sigma = 0.50$ | $\epsilon = 0.50$ | 0.63 | 0.55 | 0.47 | 0.39 | 0.30 | 0.24 | 0.19 | 0.14 | 0.00 | 0.00 |
| | $\sigma = 0.50$ | $\epsilon = 1.00$ | 0.57 | 0.52 | 0.46 | 0.41 | 0.33 | 0.28 | 0.23 | 0.18 | 0.00 | 0.00 |
| | $\sigma = 0.50$ | $\epsilon = 2.00$ | 0.47 | 0.45 | 0.41 | 0.39 | 0.35 | 0.31 | 0.26 | 0.23 | 0.00 | 0.00 |
| | $\sigma = 1.00$ | $\epsilon = 0.25$ | 0.48 | 0.42 | 0.36 | 0.30 | 0.25 | 0.21 | 0.17 | 0.14 | 0.12 | 0.09 |
| | $\sigma = 1.00$ | $\epsilon = 0.50$ | 0.47 | 0.41 | 0.37 | 0.31 | 0.27 | 0.22 | 0.20 | 0.17 | 0.14 | 0.12 |
| | $\sigma = 1.00$ | $\epsilon = 1.00$ | 0.46 | 0.41 | 0.37 | 0.33 | 0.28 | 0.24 | 0.22 | 0.18 | 0.16 | 0.14 |
| | $\sigma = 1.00$ | $\epsilon = 2.00$ | 0.39 | 0.37 | 0.34 | 0.30 | 0.27 | 0.25 | 0.22 | 0.20 | 0.18 | 0.15 |
| $m_{train} = 2$ SAMPLES | $\sigma = 0.12$ | $\epsilon = 0.25$ | 0.83 | 0.71 | 0.00 | 0.00 | 0.00 | 0.00 | 0.00 | 0.00 | 0.00 | 0.00 |
| | $\sigma = 0.12$ | $\epsilon = 0.50$ | 0.80 | 0.68 | 0.00 | 0.00 | 0.00 | 0.00 | 0.00 | 0.00 | 0.00 | 0.00 |
| | $\sigma = 0.12$ | $\epsilon = 1.00$ | 0.75 | 0.64 | 0.00 | 0.00 | 0.00 | 0.00 | 0.00 | 0.00 | 0.00 | 0.00 |
| | $\sigma = 0.12$ | $\epsilon = 2.00$ | 0.61 | 0.56 | 0.00 | 0.00 | 0.00 | 0.00 | 0.00 | 0.00 | 0.00 | 0.00 |
| | $\sigma = 0.25$ | $\epsilon = 0.25$ | 0.77 | 0.66 | 0.53 | 0.38 | 0.00 | 0.00 | 0.00 | 0.00 | 0.00 | 0.00 |
| | $\sigma = 0.25$ | $\epsilon = 0.50$ | 0.73 | 0.64 | 0.53 | 0.43 | 0.00 | 0.00 | 0.00 | 0.00 | 0.00 | 0.00 |
| | $\sigma = 0.25$ | $\epsilon = 1.00$ | 0.68 | 0.61 | 0.55 | **0.46** | 0.00 | 0.00 | 0.00 | 0.00 | 0.00 | 0.00 |
| | $\sigma = 0.25$ | $\epsilon = 2.00$ | 0.58 | 0.53 | 0.48 | 0.44 | 0.00 | 0.00 | 0.00 | 0.00 | 0.00 | 0.00 |
| | $\sigma = 0.50$ | $\epsilon = 0.25$ | 0.65 | 0.58 | 0.48 | 0.38 | 0.28 | 0.21 | 0.13 | 0.08 | 0.00 | 0.00 |
| | $\sigma = 0.50$ | $\epsilon = 0.50$ | 0.64 | 0.56 | 0.50 | 0.41 | 0.32 | 0.27 | 0.17 | 0.12 | 0.00 | 0.00 |
| | $\sigma = 0.50$ | $\epsilon = 1.00$ | 0.60 | 0.54 | 0.48 | 0.41 | 0.35 | 0.29 | 0.24 | 0.19 | 0.00 | 0.00 |
| | $\sigma = 0.50$ | $\epsilon = 2.00$ | 0.52 | 0.48 | 0.45 | 0.41 | **0.38** | **0.33** | **0.28** | **0.23** | 0.00 | 0.00 |
| | $\sigma = 1.00$ | $\epsilon = 0.25$ | 0.50 | 0.43 | 0.37 | 0.31 | 0.25 | 0.21 | 0.17 | 0.14 | 0.11 | 0.08 |
| | $\sigma = 1.00$ | $\epsilon = 0.50$ | 0.48 | 0.43 | 0.37 | 0.32 | 0.26 | 0.22 | 0.19 | 0.16 | 0.14 | 0.10 |
| | $\sigma = 1.00$ | $\epsilon = 1.00$ | 0.46 | 0.43 | 0.39 | 0.34 | 0.30 | 0.26 | 0.23 | 0.19 | 0.16 | 0.14 |
| | $\sigma = 1.00$ | $\epsilon = 2.00$ | 0.43 | 0.41 | 0.36 | 0.32 | 0.29 | 0.26 | 0.24 | 0.21 | 0.17 | 0.15 |
| $m_{train} = 4$ SAMPLES | $\sigma = 0.12$ | $\epsilon = 0.25$ | **0.86** | 0.71 | 0.00 | 0.00 | 0.00 | 0.00 | 0.00 | 0.00 | 0.00 | 0.00 |
| | $\sigma = 0.12$ | $\epsilon = 0.50$ | 0.83 | 0.70 | 0.00 | 0.00 | 0.00 | 0.00 | 0.00 | 0.00 | 0.00 | 0.00 |
| | $\sigma = 0.12$ | $\epsilon = 1.00$ | 0.77 | 0.66 | 0.00 | 0.00 | 0.00 | 0.00 | 0.00 | 0.00 | 0.00 | 0.00 |
| | $\sigma = 0.12$ | $\epsilon = 2.00$ | 0.64 | 0.58 | 0.00 | 0.00 | 0.00 | 0.00 | 0.00 | 0.00 | 0.00 | 0.00 |
| | $\sigma = 0.25$ | $\epsilon = 0.25$ | 0.79 | 0.66 | 0.52 | 0.37 | 0.00 | 0.00 | 0.00 | 0.00 | 0.00 | 0.00 |
| | $\sigma = 0.25$ | $\epsilon = 0.50$ | 0.76 | 0.66 | **0.56** | 0.42 | 0.00 | 0.00 | 0.00 | 0.00 | 0.00 | 0.00 |
| | $\sigma = 0.25$ | $\epsilon = 1.00$ | 0.72 | 0.64 | 0.56 | 0.46 | 0.00 | 0.00 | 0.00 | 0.00 | 0.00 | 0.00 |
| | $\sigma = 0.25$ | $\epsilon = 2.00$ | 0.60 | 0.56 | 0.50 | 0.44 | 0.00 | 0.00 | 0.00 | 0.00 | 0.00 | 0.00 |
| | $\sigma = 0.50$ | $\epsilon = 0.25$ | 0.67 | 0.58 | 0.47 | 0.38 | 0.27 | 0.20 | 0.13 | 0.08 | 0.00 | 0.00 |
| | $\sigma = 0.50$ | $\epsilon = 0.50$ | 0.66 | 0.59 | 0.50 | 0.39 | 0.30 | 0.24 | 0.18 | 0.11 | 0.00 | 0.00 |
| | $\sigma = 0.50$ | $\epsilon = 1.00$ | 0.63 | 0.57 | 0.49 | 0.41 | 0.36 | 0.29 | 0.23 | 0.18 | 0.00 | 0.00 |
| | $\sigma = 0.50$ | $\epsilon = 2.00$ | 0.53 | 0.49 | 0.46 | 0.41 | 0.36 | 0.32 | 0.27 | 0.21 | 0.00 | 0.00 |
| | $\sigma = 1.00$ | $\epsilon = 0.25$ | 0.50 | 0.43 | 0.38 | 0.30 | 0.25 | 0.20 | 0.16 | 0.13 | 0.10 | 0.07 |
| | $\sigma = 1.00$ | $\epsilon = 0.50$ | 0.49 | 0.44 | 0.39 | 0.32 | 0.27 | 0.23 | 0.18 | 0.15 | 0.12 | 0.09 |
| | $\sigma = 1.00$ | $\epsilon = 1.00$ | 0.49 | 0.45 | 0.41 | 0.35 | 0.29 | 0.24 | 0.19 | 0.18 | 0.15 | 0.13 |
| | $\sigma = 1.00$ | $\epsilon = 2.00$ | 0.46 | 0.42 | 0.38 | 0.35 | 0.31 | 0.28 | 0.24 | 0.21 | **0.19** | **0.16** |
| $m_{train} = 8$ SAMPLES | $\sigma = 0.12$ | $\epsilon = 0.25$ | 0.85 | 0.69 | 0.00 | 0.00 | 0.00 | 0.00 | 0.00 | 0.00 | 0.00 | 0.00 |
| | $\sigma = 0.12$ | $\epsilon = 0.50$ | 0.84 | **0.72** | 0.00 | 0.00 | 0.00 | 0.00 | 0.00 | 0.00 | 0.00 | 0.00 |
| | $\sigma = 0.12$ | $\epsilon = 1.00$ | 0.80 | 0.71 | 0.00 | 0.00 | 0.00 | 0.00 | 0.00 | 0.00 | 0.00 | 0.00 |
| | $\sigma = 0.12$ | $\epsilon = 2.00$ | 0.66 | 0.59 | 0.00 | 0.00 | 0.00 | 0.00 | 0.00 | 0.00 | 0.00 | 0.00 |
| | $\sigma = 0.25$ | $\epsilon = 0.25$ | 0.81 | 0.67 | 0.52 | 0.40 | 0.00 | 0.00 | 0.00 | 0.00 | 0.00 | 0.00 |
| | $\sigma = 0.25$ | $\epsilon = 0.50$ | 0.10 | 0.10 | 0.10 | 0.10 | 0.00 | 0.00 | 0.00 | 0.00 | 0.00 | 0.00 |
| | $\sigma = 0.25$ | $\epsilon = 1.00$ | 0.73 | 0.65 | 0.56 | 0.46 | 0.00 | 0.00 | 0.00 | 0.00 | 0.00 | 0.00 |
| | $\sigma = 0.25$ | $\epsilon = 2.00$ | 0.61 | 0.56 | 0.51 | 0.45 | 0.00 | 0.00 | 0.00 | 0.00 | 0.00 | 0.00 |
| | $\sigma = 0.50$ | $\epsilon = 0.25$ | 0.67 | 0.57 | 0.47 | 0.37 | 0.28 | 0.19 | 0.13 | 0.08 | 0.00 | 0.00 |
| | $\sigma = 0.50$ | $\epsilon = 0.50$ | 0.68 | 0.59 | 0.48 | 0.41 | 0.30 | 0.22 | 0.16 | 0.11 | 0.00 | 0.00 |
| | $\sigma = 0.50$ | $\epsilon = 1.00$ | 0.62 | 0.56 | 0.50 | 0.42 | 0.35 | 0.29 | 0.20 | 0.14 | 0.00 | 0.00 |
| | $\sigma = 0.50$ | $\epsilon = 2.00$ | 0.57 | 0.51 | 0.46 | 0.42 | 0.38 | 0.32 | 0.28 | 0.22 | 0.00 | 0.00 |
| | $\sigma = 1.00$ | $\epsilon = 0.25$ | 0.49 | 0.43 | 0.36 | 0.30 | 0.26 | 0.19 | 0.15 | 0.13 | 0.10 | 0.07 |
| | $\sigma = 1.00$ | $\epsilon = 0.50$ | 0.48 | 0.41 | 0.35 | 0.28 | 0.21 | 0.17 | 0.15 | 0.11 | 0.08 | 0.05 |
| | $\sigma = 1.00$ | $\epsilon = 1.00$ | 0.46 | 0.42 | 0.38 | 0.33 | 0.27 | 0.24 | 0.20 | 0.16 | 0.15 | 0.12 |
| | $\sigma = 1.00$ | $\epsilon = 2.00$ | 0.47 | 0.44 | 0.40 | 0.37 | 0.32 | 0.27 | 0.23 | 0.20 | 0.17 | 0.15 |

Table 15: SMOOTHADV$_{\text{PGD}}$ training $T = 10$ steps, $m_{train} \in \{1, 2, 4, 8\}$ samples.

| | $\ell_2$ RADIUS (CIFAR-10) | 0.0 | 0.25 | 0.5 | 0.75 | 1.0 | 1.25 | 1.5 | 1.75 | 2.0 | 2.25 |
|---|---|---|---|---|---|---|---|---|---|---|---|
| [6] | $\sigma = 0.12$ | 0.81 | 0.59 | 0.00 | 0.00 | 0.00 | 0.00 | 0.00 | 0.00 | 0.00 | 0.00 |
| | $\sigma = 0.25$ | 0.75 | 0.60 | 0.43 | 0.27 | 0.00 | 0.00 | 0.00 | 0.00 | 0.00 | 0.00 |
| | $\sigma = 0.50$ | 0.65 | 0.55 | 0.41 | 0.32 | 0.23 | 0.15 | 0.09 | 0.05 | 0.00 | 0.00 |
| | $\sigma = 1.00$ | 0.47 | 0.39 | 0.34 | 0.28 | 0.22 | 0.17 | 0.14 | 0.12 | 0.10 | 0.08 |
| $m_{train} = 1$ SAMPLE | $\sigma = 0.12$ $\epsilon = 0.25$ | 0.82 | 0.69 | 0.00 | 0.00 | 0.00 | 0.00 | 0.00 | 0.00 | 0.00 | 0.00 |
| | $\sigma = 0.12$ $\epsilon = 0.50$ | 0.79 | 0.68 | 0.00 | 0.00 | 0.00 | 0.00 | 0.00 | 0.00 | 0.00 | 0.00 |
| | $\sigma = 0.12$ $\epsilon = 1.00$ | 0.72 | 0.62 | 0.00 | 0.00 | 0.00 | 0.00 | 0.00 | 0.00 | 0.00 | 0.00 |
| | $\sigma = 0.12$ $\epsilon = 2.00$ | 0.54 | 0.52 | 0.00 | 0.00 | 0.00 | 0.00 | 0.00 | 0.00 | 0.00 | 0.00 |
| | $\sigma = 0.25$ $\epsilon = 0.25$ | 0.74 | 0.63 | 0.50 | 0.39 | 0.00 | 0.00 | 0.00 | 0.00 | 0.00 | 0.00 |
| | $\sigma = 0.25$ $\epsilon = 0.50$ | 0.75 | 0.64 | 0.53 | 0.43 | 0.00 | 0.00 | 0.00 | 0.00 | 0.00 | 0.00 |
| | $\sigma = 0.25$ $\epsilon = 1.00$ | 0.63 | 0.56 | 0.51 | 0.46 | 0.00 | 0.00 | 0.00 | 0.00 | 0.00 | 0.00 |
| | $\sigma = 0.25$ $\epsilon = 2.00$ | 0.53 | 0.49 | 0.46 | 0.42 | 0.00 | 0.00 | 0.00 | 0.00 | 0.00 | 0.00 |
| | $\sigma = 0.50$ $\epsilon = 0.25$ | 0.62 | 0.54 | 0.44 | 0.36 | 0.27 | 0.22 | 0.14 | 0.10 | 0.00 | 0.00 |
| | $\sigma = 0.50$ $\epsilon = 0.50$ | 0.63 | 0.54 | 0.45 | 0.37 | 0.31 | 0.26 | 0.19 | 0.13 | 0.00 | 0.00 |
| | $\sigma = 0.50$ $\epsilon = 1.00$ | 0.55 | 0.51 | 0.45 | 0.38 | 0.33 | 0.28 | 0.24 | 0.18 | 0.00 | 0.00 |
| | $\sigma = 0.50$ $\epsilon = 2.00$ | 0.49 | 0.46 | 0.42 | 0.37 | 0.35 | 0.31 | 0.26 | **0.23** | 0.00 | 0.00 |
| | $\sigma = 1.00$ $\epsilon = 0.25$ | 0.49 | 0.41 | 0.36 | 0.31 | 0.25 | 0.22 | 0.18 | 0.15 | 0.12 | 0.10 |
| | $\sigma = 1.00$ $\epsilon = 0.50$ | 0.47 | 0.41 | 0.36 | 0.31 | 0.27 | 0.23 | 0.20 | 0.16 | 0.14 | 0.11 |
| | $\sigma = 1.00$ $\epsilon = 1.00$ | 0.46 | 0.43 | 0.38 | 0.33 | 0.29 | 0.26 | 0.22 | 0.18 | 0.15 | 0.14 |
| | $\sigma = 1.00$ $\epsilon = 2.00$ | 0.40 | 0.36 | 0.34 | 0.32 | 0.28 | 0.24 | 0.22 | 0.20 | 0.18 | 0.16 |
| $m_{train} = 2$ SAMPLES | $\sigma = 0.12$ $\epsilon = 0.25$ | 0.83 | 0.71 | 0.00 | 0.00 | 0.00 | 0.00 | 0.00 | 0.00 | 0.00 | 0.00 |
| | $\sigma = 0.12$ $\epsilon = 0.50$ | 0.82 | 0.70 | 0.00 | 0.00 | 0.00 | 0.00 | 0.00 | 0.00 | 0.00 | 0.00 |
| | $\sigma = 0.12$ $\epsilon = 1.00$ | 0.74 | 0.66 | 0.00 | 0.00 | 0.00 | 0.00 | 0.00 | 0.00 | 0.00 | 0.00 |
| | $\sigma = 0.12$ $\epsilon = 2.00$ | 0.60 | 0.54 | 0.00 | 0.00 | 0.00 | 0.00 | 0.00 | 0.00 | 0.00 | 0.00 |
| | $\sigma = 0.25$ $\epsilon = 0.25$ | 0.77 | 0.64 | 0.50 | 0.38 | 0.00 | 0.00 | 0.00 | 0.00 | 0.00 | 0.00 |
| | $\sigma = 0.25$ $\epsilon = 0.50$ | 0.73 | 0.65 | 0.53 | 0.44 | 0.00 | 0.00 | 0.00 | 0.00 | 0.00 | 0.00 |
| | $\sigma = 0.25$ $\epsilon = 1.00$ | 0.70 | 0.61 | 0.54 | 0.46 | 0.00 | 0.00 | 0.00 | 0.00 | 0.00 | 0.00 |
| | $\sigma = 0.25$ $\epsilon = 2.00$ | 0.57 | 0.53 | 0.51 | 0.45 | 0.00 | 0.00 | 0.00 | 0.00 | 0.00 | 0.00 |
| | $\sigma = 0.50$ $\epsilon = 0.25$ | 0.66 | 0.55 | 0.45 | 0.37 | 0.30 | 0.21 | 0.14 | 0.09 | 0.00 | 0.00 |
| | $\sigma = 0.50$ $\epsilon = 0.50$ | 0.64 | 0.56 | 0.49 | 0.40 | 0.32 | 0.25 | 0.16 | 0.11 | 0.00 | 0.00 |
| | $\sigma = 0.50$ $\epsilon = 1.00$ | 0.61 | 0.55 | 0.48 | 0.41 | 0.35 | 0.28 | 0.22 | 0.17 | 0.00 | 0.00 |
| | $\sigma = 0.50$ $\epsilon = 2.00$ | 0.50 | 0.46 | 0.44 | 0.40 | **0.38** | **0.33** | **0.29** | 0.23 | 0.00 | 0.00 |
| | $\sigma = 1.00$ $\epsilon = 0.25$ | 0.50 | 0.44 | 0.38 | 0.31 | 0.26 | 0.21 | 0.17 | 0.15 | 0.11 | 0.09 |
| | $\sigma = 1.00$ $\epsilon = 0.50$ | 0.48 | 0.44 | 0.39 | 0.33 | 0.27 | 0.22 | 0.19 | 0.16 | 0.13 | 0.10 |
| | $\sigma = 1.00$ $\epsilon = 1.00$ | 0.47 | 0.44 | 0.39 | 0.35 | 0.30 | 0.25 | 0.21 | 0.18 | 0.16 | 0.14 |
| | $\sigma = 1.00$ $\epsilon = 2.00$ | 0.45 | 0.41 | 0.38 | 0.35 | 0.32 | 0.28 | 0.25 | 0.22 | **0.19** | **0.17** |
| $m_{train} = 4$ SAMPLES | $\sigma = 0.12$ $\epsilon = 0.25$ | 0.85 | **0.74** | 0.00 | 0.00 | 0.00 | 0.00 | 0.00 | 0.00 | 0.00 | 0.00 |
| | $\sigma = 0.12$ $\epsilon = 0.50$ | 0.80 | 0.68 | 0.00 | 0.00 | 0.00 | 0.00 | 0.00 | 0.00 | 0.00 | 0.00 |
| | $\sigma = 0.12$ $\epsilon = 1.00$ | 0.75 | 0.69 | 0.00 | 0.00 | 0.00 | 0.00 | 0.00 | 0.00 | 0.00 | 0.00 |
| | $\sigma = 0.12$ $\epsilon = 2.00$ | 0.61 | 0.57 | 0.00 | 0.00 | 0.00 | 0.00 | 0.00 | 0.00 | 0.00 | 0.00 |
| | $\sigma = 0.25$ $\epsilon = 0.25$ | 0.78 | 0.64 | 0.50 | 0.37 | 0.00 | 0.00 | 0.00 | 0.00 | 0.00 | 0.00 |
| | $\sigma = 0.25$ $\epsilon = 0.50$ | 0.78 | 0.68 | 0.53 | 0.41 | 0.00 | 0.00 | 0.00 | 0.00 | 0.00 | 0.00 |
| | $\sigma = 0.25$ $\epsilon = 1.00$ | 0.72 | 0.65 | 0.56 | **0.48** | 0.00 | 0.00 | 0.00 | 0.00 | 0.00 | 0.00 |
| | $\sigma = 0.25$ $\epsilon = 2.00$ | 0.61 | 0.56 | 0.51 | 0.47 | 0.00 | 0.00 | 0.00 | 0.00 | 0.00 | 0.00 |
| | $\sigma = 0.50$ $\epsilon = 0.25$ | 0.67 | 0.59 | 0.45 | 0.34 | 0.28 | 0.18 | 0.13 | 0.08 | 0.00 | 0.00 |
| | $\sigma = 0.50$ $\epsilon = 0.50$ | 0.66 | 0.56 | 0.48 | 0.38 | 0.29 | 0.22 | 0.16 | 0.10 | 0.00 | 0.00 |
| | $\sigma = 0.50$ $\epsilon = 1.00$ | 0.64 | 0.57 | 0.49 | 0.40 | 0.33 | 0.28 | 0.22 | 0.16 | 0.00 | 0.00 |
| | $\sigma = 0.50$ $\epsilon = 2.00$ | 0.53 | 0.50 | 0.45 | 0.41 | 0.35 | 0.30 | 0.27 | 0.23 | 0.00 | 0.00 |
| | $\sigma = 1.00$ $\epsilon = 0.25$ | 0.49 | 0.43 | 0.37 | 0.29 | 0.25 | 0.21 | 0.16 | 0.13 | 0.10 | 0.08 |
| | $\sigma = 1.00$ $\epsilon = 0.50$ | 0.50 | 0.43 | 0.37 | 0.30 | 0.26 | 0.21 | 0.17 | 0.13 | 0.11 | 0.09 |
| | $\sigma = 1.00$ $\epsilon = 1.00$ | 0.10 | 0.10 | 0.10 | 0.10 | 0.10 | 0.10 | 0.10 | 0.10 | 0.10 | 0.10 |
| | $\sigma = 1.00$ $\epsilon = 2.00$ | 0.45 | 0.43 | 0.38 | 0.36 | 0.33 | 0.30 | 0.25 | 0.22 | 0.19 | 0.16 |
| $m_{train} = 8$ SAMPLES | $\sigma = 0.12$ $\epsilon = 0.25$ | **0.86** | 0.71 | 0.00 | 0.00 | 0.00 | 0.00 | 0.00 | 0.00 | 0.00 | 0.00 |
| | $\sigma = 0.12$ $\epsilon = 0.50$ | 0.83 | 0.71 | 0.00 | 0.00 | 0.00 | 0.00 | 0.00 | 0.00 | 0.00 | 0.00 |
| | $\sigma = 0.12$ $\epsilon = 1.00$ | 0.78 | 0.69 | 0.00 | 0.00 | 0.00 | 0.00 | 0.00 | 0.00 | 0.00 | 0.00 |
| | $\sigma = 0.12$ $\epsilon = 2.00$ | 0.63 | 0.60 | 0.00 | 0.00 | 0.00 | 0.00 | 0.00 | 0.00 | 0.00 | 0.00 |
| | $\sigma = 0.25$ $\epsilon = 0.25$ | 0.82 | 0.68 | 0.52 | 0.37 | 0.00 | 0.00 | 0.00 | 0.00 | 0.00 | 0.00 |
| | $\sigma = 0.25$ $\epsilon = 0.50$ | 0.76 | 0.65 | 0.53 | 0.41 | 0.00 | 0.00 | 0.00 | 0.00 | 0.00 | 0.00 |
| | $\sigma = 0.25$ $\epsilon = 1.00$ | 0.74 | 0.67 | **0.57** | 0.47 | 0.00 | 0.00 | 0.00 | 0.00 | 0.00 | 0.00 |
| | $\sigma = 0.25$ $\epsilon = 2.00$ | 0.61 | 0.57 | 0.53 | 0.46 | 0.00 | 0.00 | 0.00 | 0.00 | 0.00 | 0.00 |
| | $\sigma = 0.50$ $\epsilon = 0.25$ | 0.68 | 0.58 | 0.48 | 0.37 | 0.27 | 0.17 | 0.10 | 0.07 | 0.00 | 0.00 |
| | $\sigma = 0.50$ $\epsilon = 0.50$ | 0.66 | 0.58 | 0.48 | 0.38 | 0.29 | 0.22 | 0.16 | 0.10 | 0.00 | 0.00 |
| | $\sigma = 0.50$ $\epsilon = 1.00$ | 0.64 | 0.57 | 0.49 | 0.41 | 0.34 | 0.27 | 0.22 | 0.14 | 0.00 | 0.00 |
| | $\sigma = 0.50$ $\epsilon = 2.00$ | 0.09 | 0.09 | 0.09 | 0.09 | 0.09 | 0.09 | 0.09 | 0.09 | 0.00 | 0.00 |
| | $\sigma = 1.00$ $\epsilon = 0.25$ | 0.48 | 0.40 | 0.34 | 0.29 | 0.23 | 0.17 | 0.14 | 0.11 | 0.07 | 0.05 |
| | $\sigma = 1.00$ $\epsilon = 0.50$ | 0.48 | 0.44 | 0.37 | 0.32 | 0.27 | 0.22 | 0.19 | 0.16 | 0.13 | 0.09 |
| | $\sigma = 1.00$ $\epsilon = 1.00$ | 0.50 | 0.44 | 0.38 | 0.34 | 0.29 | 0.24 | 0.20 | 0.16 | 0.13 | 0.10 |
| | $\sigma = 1.00$ $\epsilon = 2.00$ | 0.46 | 0.43 | 0.40 | 0.36 | 0.30 | 0.29 | 0.24 | 0.20 | 0.18 | 0.15 |

Table 16: Certified top-1 accuracy of our best ImageNet classifiers (over a subsample of 500 points) at various $\ell_2$ radii. Standard accuracies are in parantheses.

| $\ell_2$ RADIUS (IMAGENET) | 0.5 | 1.0 | 1.5 | 2.0 | 2.5 | 3.0 | 3.5 |
|---|---|---|---|---|---|---|---|
| COHEN ET AL. [6] (%) | (67)49 | (57)37 | (57)29 | (44)19 | (44)15 | (44)12 | (44)9 |
| OURS (%) | (65)**56** | (55)**45** | (55)**38** | (42)**28** | (42)**26** | (42)**20** | (35)**17** |

Table 17: Certified top-1 accuracy of our best CIFAR-10 classifiers (on the full test set) at various $\ell_2$ radii. Standard accuracies are in parantheses.

| $\ell_2$ RADIUS (CIFAR-10) | 0.25 | 0.5 | 0.75 | 1.0 | 1.25 | 1.5 | 1.75 | 2.0 | 2.25 |
|---|---|---|---|---|---|---|---|---|---|
| COHEN ET AL. [6] (%) | (77)61 | (66)43 | (66)32 | (47)22 | (47)17 | (47)13 | (47)10 | (47)7 | (47)4 |
| OURS (%) | (85)73 | (76)58 | (75)48 | (57)38 | (53)33 | (53)29 | (53)24 | (44)18 | (44)16 |
| + PRE-TRAINING (%) | (90)80 | (81)62 | (74)**52** | (54)38 | (54)**34** | (54)**30** | (54)**25** | (41)**19** | (41)16 |
| + SEMI-SUPERVISION (%) | (90)80 | (80)**63** | (80)**52** | (62)**40** | (62)**34** | (52)29 | (52)**25** | (42)**19** | (42)**17** |
| + BOTH(%) | (91)**81** | (81)**63** | (72)**52** | (51)37 | (51)33 | (51)29 | (51)**25** | (41)18 | (41)16 |

## Footnotes

[7]The 500K unlabelled dataset was not public at the time this paper was written. We obtained it, along with the pseudo-labels, from the authors of [5]. We refer the reader to the authors of [5] to obtain this dataset if interested in replicating our self-training results.