[Reviews · NeurIPS 2019]

Reviewer 1



Originality: This work combines existing techniques in a meaningful way. The attack is very closely related to the Expectation over Transformation attack in "Synthesizing Robust Adversarial Examples" by Athalye et al(which needs to be cited). It is already established that stronger attacks can help in adversarial training, and the authors verify this in their experiments. The authors also attempt to use Stein's lemma to get a better estimate of the gradient, although they claim it does not help empirically. Quality: The authors experimentally validate all the claims they make, and the paper is technically sound. Clarity: I think the paper is clearly written and organized, and the code can be reproduced by other researchers. Significance: Although this submission borrows most of its technical content from other works, I think it is significant because it improves the empirical and certifiable robustness of Cohen at al. The authors decrease the empirical robustness of Cohen et al via their attack, but there is still a big gap between the certifiable robustness and the empirical robustness on Cohen et al. Miscellaneous: It would help to change the scale of accuracy plotted in the figures. For example, in Figure 3, at L2=0.5, the empirical accuracy against your attacks looks similar to vanilla PGD, simply because 50% and 62.5% are very close to each other in the plot. ---Edit after rebuttal--- After reading the rebuttal and other reviews, I have increased my score to 7. I accept your argument that your modification from Athalye et al is significant and may look obvious in hindsight. Although I did not criticize the theory in my original review, I agree that you should include your alternative proof.

Reviewer 2



Overall, I believe the paper makes a meaningful empirical contribution to scalable training methods of robust classifiers. By finding adversarial examples for smoothed classifiers and modifying the training procedure, the authors significantly improve the accuracy of smoothed classifiers. Smoothed classifiers are of interest since they are scalable and come with a certificate of robustness. The paper is clearly written. However, the contribution seems incremental. All the building blocks have been introduced in the past: adversarial training, smoothed classifiers, robustness of smoothed classifiers. The only contribution is the method for finding adversarial examples for smoothed classifiers, which is interesting, but I would have expected more depth. The paper would be significant if they had a theorem that somehow argued that robustly trained smoothed classifiers were somehow optimal in a stronger sense. Detailed comments. Is SmoothedAdv different in any way from the standard robust training for appropriate attacks (Madry et al 2017)? Lines 33-34. "... to substantially improve the previous certified robustness results of randomized smoothing". To me, this suggests that the contribution is theoretical, and the theoretical robustness guarantees are improved. However, the contribution is only empirical. It would be appreciated if the authors could make this clear to the reader. Lines 118-121. A similar comment as above. The authors don't show that adversarial training improves the robust accuracy of the base or the smoothed classifier. They only observe it empirically for the smoothed classifier, which already comes with a certificate. Section 2.1 and Experiments. How many monte carlo samples one would need to get high probability estimates of lower/upper bounds on p_A, p_B? What's the probability of your robustness guarantee (for the parameter choices used in the experiments)? (*) Experiments section: Only the accuracies on examples, where the smoothed classifier made predictions (did not abstain) are plotted. The comparison that seems to be missing is the fraction of examples on which the two smoothed classifiers abstained. If the number is similar, is it always the same set of examples? Lines 246-247. It would be nice if the authors could add a sentence or two summarizing the results of this experiment, so that the reader would not have to search for it in the appendix. ******* Update after Author Response: After reading other reviews and author response, I have raised my score to 7. While all the ideas seem incremental, overall the work seems to make a significant contribution. I would also like to add that my score increase is based on the following promised paper improvements made by the authors in their response: - "Our main contribution is not theoretical and we will update the draft to make this more clear." - "include an alternate proof of the tight certified bound of smoothed classifiers ".

Reviewer 3



I believe this is an important, significant contribution which warrants acceptance at NeurIPS. While the idea to combine Randomized Smoothing with Adversarial Training may appear straight-forward, the authors had to address a number of technical challenges (like designing a PGD variant to work on randomly smoothed classifier) which they did in a very careful, systematic and clear way. The Appendix documents a lot of the directions that the authors explored along the way. One suggestion: the performance of the new state-of-the-art certified classifiers on clean samples is a bit hidden (one has to look very closely at Figures 1 and 2). If space permits, I think it would be good to include them in Table 1 & 2 (Cohen et al. had highlighted the standard accuracy of their certified classifiers in such a way). One related question: what was the procedure for picking the "representative" models in Figure 1 and 2? And one minor comment: l.213: "see for explanation" ---------------------------------------- I've read the authors' response which confirmed my very positive impression of this submission.

Reviewer 4



Clarity: This work is clearly written and easy to follow. Quality: The experimental section is very thorough and all steps are explained very clearly. The figures comparing against the Cohen paper is familiar, and therefore easy to parse, and seemingly very fair. Significance + Originality: (theory) While I feel that the thread on randomized smoothing is an incredibly promising and interesting research direction towards certifiably robust classifiers, this work hardly serves as a standalone result, instead reading like a sequel to the Cohen paper. Keeping in mind that the primary contribution of Cohen's work was the tighter analysis leveraging Neymon-Pearson (indeed, Lecuyer + Li both incorporated AWGN into their smoothed classifiers), this work provides no such analysis or new theoretical insight. Caveat: appendix B does provide an alternative perspective for deriving smoothAdv, but it does not seem to be utilized anywhere. Further, it is natural to apply randomized smoothing to soft classifiers and the corrected objective function (equation (S)) seems simply like correcting a bug from Cohen's work. The application of Stein's lemma to derive an unbiased estimator and the alternative derivation presented in appendix B is perhaps promising, but ultimately not useful. (practice): the main contribution of this work is that it provides state of the art empirical results for robustness against L2 perturbations smoothed classifiers, substantially improving upon the numbers put forth by Cohen et al. The numerical improvements are undeniable, yet not particularly interesting. The objective used in creating adversarial examples is very natural, and certainly it stands to reason that adversarial training will improve the robust accuracy (and certifiable radius) of a classifier. (some minor typos): - the middle term in eq. (7) is lacking a subscript y - pseudocode 2 defines counts twice (the first counts should be counts0 if the authors desire to be faithful to Cohen et al). This may be confusing to readers unfamiliar with the original work. ################################################################## UPDATE AFTER REBUTTAL: I've changed my score to 7 after reading the authors' response. I feel an alternate proof of the key result of Cohen et. al that argues for stronger robustness via reduction of the lipschitz constant provides a nice intuition towards how and why this technique works. This is the primary reason why I've chosen to increase my score.

[Author Response · NeurIPS 2019]

1 We thank the reviewers for their comments. Before we address individual concerns, we make some general comments.

2 1) We agree that our conceptual contribution can be viewed as "taking the right perspective on the problem". In hindsight this perspective was clearly the right thing to do. However we would like to point out that both previous papers (Lecuyer et al. and Cohen et al.) made the same mistake, indicating that this "right perspective" was perhaps not obvious a priori (and indeed, as we explain in Section 2.2, the previously used objective (4) also has a natural interpretation, albeit not the correct one for the problem at hand). In fact, driving this point home, Reviewer 1 cites yet another work (Athalye et al.) which uses the suboptimal objective (4) instead of our proposed objective (S).

8 2) We would like to emphasize a key point from reviewer 3: "the authors had to address a number of technical challenge [...] which they did in a very careful, systematic and clear way". Indeed, writing down the correct objective is only the first step in creating an efficient and state of the art deep learning system. Our double digit improvements over the previous state of the art numbers from Cohen et al. speak for themselves: this can only happen by getting all parts of the system right (in the present case this includes for example using a \*biased\* estimator of the gradient, see (6) and the discussion after).

14 3) Finally several reviewers ask about our theoretical insights. As reviewer 4 points out, the Stein's lemma observation sounds promising, and we leave it as an open avenue for future works. We also obtained another new theoretical result, which we left out from the submission as we wanted to focus more on the practical implications of our work (we we believe is the SOTA certified accuracy). This new insight is an alternative, and much simpler, proof of Cohen et al.'s key theorem, which relies on rephrasing Cohen et al.'s statement as a nonlinear Lipschitz bound on the smoothed classifier. This also opens up another avenue for future work, namely by finding better nonlinear Lipschitz guarantees. We would be happy to include this derivation if the reviewers think it would improve the paper.

21 ***Reviewer 1:*** We will change the scale of accuracy plotted in Figure 3, to better illustrate the difference between the empirical accuracy against SmoothAdv and the vanilla PGD.

23 As mentioned above, the Expectation over Transformation attack of Athalye et al. has the opposite order of log and expectation compared to our SmoothAdv object (Eqn. (S)), and identifying this correct order is a key contribution of our paper (see Sec 2.2). We will add this citation and a discussion of this interesting connection.

26 ***Reviewer 2:*** *Possible confusion concerning the abstention rate.* The reported certifiably robust accuracies in our paper follows the formula $\frac{\text{certified robust}}{\text{total samples}} = \frac{\text{certified robust}}{\text{certified robust+not certified robust+abstained}}$, where abstained samples are counted toward the denominator. Thus, high abstention rate leads to lower certified accuracies. Given that our certified accuracies are higher than Cohen et al.'s (who reports abstention rate of 1% with the parameters described below; see their appendix D2), our abstention rate is not likely to be lower, and it should not be the major component of our improvement over their results. Nevertheless, we will add this experiment to the final version of the paper.

32 *SmoothAdv vs PGD Training.* We emphasize that SmoothAdv is first and foremost an objective function that we argue is the correct one for attacking smoothed classifiers (see Eqn.(S)). It is thus somewhat orthogonal to "PGD training" as the objective SmoothAdv can be empirically optimized by PGD, DDN, or other attacks (see Pseudocode 1), and it can be used in both adversarial attack and adversarial training. If the reviewer is referring to adversarial training using SmoothAdv objective (with PGD optimizer), this is different from PGD training, as it carefully combines Gaussian data augmentation and adversarial training (Pseudocode 1).

38 *Our main contribution is not theoretical and we will update the draft to make this more clear.*

39 *We will summarize the experiments referred to in Lines 246-247.*

40 ***Reviewer 3:*** We thank Reviewer 3 for recognizing our careful experimental work and our technical contributions to the robust training of smoothed classifiers. We will include the standard accuracy in Table 1. The "representative" models in Figures 1 and 2 were picked at random from the set of models we train.

43 ***Reviewer 4:*** *Suggested Improvement 1, new theoretical perspective.* As mentioned in general comment (3) above, we would like to include an alternate proof of the tight certified bound of smoothed classifiers (Theorem 1 in Cohen et al.) purely by bounding the Lipschitz constant of $\Phi^{-1}(p_c)$, where $p_c$ is the probability of predicting class $c$ by the smoothed classifier and $\Phi^{-1}$ is the inverse of the standard Gaussian CDF. This greatly simplifies the proof and provide new intuitions on how or why smoothing tends to improve the robustness under L2 perturbations: it reduces the Lipschitz constant of the classifier.

49 *Suggested Improvement 2, improved results that leverage theoretical contributions.* It is promising to design stronger attacks to smoothed classifiers based on our alternative derivation of SmoothAdv in Appendix B. This is on-going work.

[Meta-Review · NeurIPS 2019]

This work shows how to improve the previous state of the art for L2 robustness using smoothed classifiers (introduced by Cohen et al.) The empirical results are very strong in a very competitive area where many research groups are competing. The theoretical work, the presentation and the various technical details involved in using smoothness in PGD are all great contributions. This is an important paper in the space of adversarial ML.